# OsMAPK6 phosphorylation and CLG1 ubiquitylation of GW6a non-additively enhance rice grain size through stabilization of the substrate

Chen Bai[1,2,3,8], Gao-Jie Wang[1,2,3,8], Xiao-Hui Feng[1,2,3,8], Qiong Gao[1,2,8], Wei-Qing Wang[1,2], Ran Xu [4,5], Su-Jie Guo[1,2,3], Shao-Yan Shen[1,2,3], Ming Ma[1,2,3], Wen-Hui Lin [1,3], Chun-Ming Liu[1,2,3], Yunhai Li[3,6,7] & Xian-Jun Song [1,2,3,7] ✉

The chromatin modifier GRAIN WEIGHT 6a (GW6a) enhances rice grain size and yield. However, little is known about its gene network determining grain size. Here, we report that MITOGEN-ACTIVED PROTEIN KINASE 6 (OsMAPK6) and E3 ligase CHANG LI GENG 1 (CLG1) interact with and target GW6a for phosphorylation and ubiquitylation, respectively. Unexpectedly, however, in vitro and in vivo assays reveal that both of the two post-translational modifications stabilize GW6a. Furthermore, we uncover two major GW6a phosphorylation sites (serine142 and threonine186) targeted by OsMAPK6 serving an important role in modulating grain size. In addition, our genetic and molecular results suggest that the *OsMAPK6-GW6a* and *CLG1-GW6a* axes are crucial and operate in a non-additive manner to control grain size. Overall, our findings identify a previously unknown mechanism by which phosphorylation and ubiquitylation non-additively stabilize GW6a to enhance grain size, and reveal correlations and interactions of these posttranslational modifications during rice grain development.

For rice (*Oryza sativa*), a major cereal crop worldwide, seed size is a crucial component trait of grain quality and yield. In the past two decades, many genetic factors determining grain size in rice have been identified, which were categorized into several pathways such as those mediated by plant hormones, ubiquitin proteasome, mitogen-activated protein kinase (MAPK), G proteins and transcriptional regulators[1,2]. In particular, MAPK cascades are highly conserved signaling modules in many species across eukaryotes that coordinate environmental stimuli and developmental inputs to alter substrate activities via the sequential phosphorylation: MAKKKs (MAPK kinase kinases) phosphorylate and activate downstream MAPKKs (MAPK kinases), which in turn phosphorylate and activate MAPKs[3–5]. It is worth noting that a genetically defined OsMKKK10-OsMKK4-OsMAPK6 pathway has recently been reported to play an important role in the regulation of rice grain size, in which each of the three components positively regulates grain size by promoting cell proliferation in spikelet hulls[6–9]. Furthermore, one regulator of the MAPK cascade is MAPK PHOSPHATASE 1 (GRAIN SIZE AND NUMBER

[1]Key Laboratory of Plant Molecular Physiology, Institute of Botany, Chinese Academy of Sciences, Beijing 100093, China. [2]China National Botanical Garden, Beijing 100093, China. [3]College of Advanced Agricultural Sciences, University of Chinese Academy of Sciences, Beijing 100049, China. [4]Sanya Nanfan Research, Institute of Hainan University, Hainan Yazhou Bay Seed Laboratory, Sanya 572025, China. [5]College of Tropical Crops Hainan University, Hainan University, Haikou 570288, China. [6]State Key Laboratory of Plant Cell and Chromosome Engineering, Institute of Genetics and Developmental Biology, Chinese Academy of Sciences, Beijing 100101, China. [7]The Innovative Academy of Seed Design, Chinese Academy of Sciences, Beijing 100101, China. [8]These authors contributed equally: Chen Bai, Gao-Jie Wang, Xiao-Hui Feng, Qiong Gao. ✉e-mail: songxj@ibcas.ac.cn

1, GSN1), which interacts with and deactivates OsMAPK6 via dephosphorylation to negatively regulate grain size[10]. The Rho-family GTPase OsRac1 could positively regulate grain size through enhancement of phosphorylation and stabilization of OsMAPK6[11]. In addition, OsWRKY53 is a direct substrate of the OsMKKK10-OsMKK4-OsMAPK6 pathway, in which OsMAPK6 phosphorylates OsWRKY53 to increase its transcriptional activity to function in the control of grain size[12–15]. These findings greatly advanced our understanding of the MAPK pathway in the regulation of grain size; however, to date only a few of direct MAPK substrates that are involved in controlling the trait have been reported.

Meanwhile, ubiquitylation is well-known for targeting cellular proteins for the 26S proteasome-mediated degradation and homeostasis, in addition to its other functions such as modulation of protein interactions, subcellular distribution, transcription, DNA repair and propagation of transmembrane signaling, which has been linked to almost every cellular process[16–18]. The modification occurs through the sequential action of three types of protein: ubiquitin-activating enzymes (E1s), ubiquitin-conjugating enzymes (E2s), and ubiquitin ligase (E3s)[18,19]. Interestingly, several E3-participated pathways have been identified in controlling rice grain size; for example, the E3 ubiquitin ligase GRAIN WIDTH ON CHROMOSOME 2 (GW2) ubiquitinates and destabilizes the glutaredoxin protein WIDE GRAIN 1 (WG1) that interacts with and represses the transcriptional activity of OsbZIP47 by recruiting the transcriptional co-repressor ABERRANT SPIKELET AND PANICLE 1 (ASP1)[20,21]. Another significant E3 ligase CHANG LI GENG 1 (CLG1, also called HAEMERYTHRIN MOTIF-CONTAINING RING-AND ZINC-FINGER PROTEIN 2, OsHRZ2) targets the Gγ protein GRAIN SIZE 3 (GS3) for ubiquitylation and subsequent degradation through the endosome pathway to control rice grain size[22,23]. Similarly, recent studies suggested that the E3 ubiquitin ligase DECREASED GRAIN SIZE 1 (DGS1) interacts with and ubiquinates the brassinosteroid receptor BRASSINOSTEROID INSENSITIVE 1 (BRI1) facilitating its endoplasmic reticulum-associated degradation to increase grain size[24,25]. In addition, our recent finding has revealed that the ubiquitin receptor HOMOLOG OF DA1 ON RICE CHROMOSOME 3 (HDR3) increases the K63-linked ubiquitylation of and stabilizes GRAIN WEIGHT 6a (GW6a) to positively regulate rice grain size[26], although the corresponding E3 ubiquitin ligase(s) that collaborate(s) with HDR3 has still remained elusive.

*GW6a* is a positive modulator of grain size and yield, which encodes a histone H4 acetyltransferase (chromatin modifier), functioning presumably through the regulation of transcription[27]. However, its gene network controlling grain size has remained largely unknown. Here, we coincidentally identified a mitogen-activated protein kinase (OsMAPK6) and a RING-type E3 ligase (CLG1) as another two GW6a interacting proteins in a yeast two-hybrid screening assay. Furthermore, our in vitro and in vivo biochemical evidence suggested that OsMAPK6 and CLG1 targeted GW6a, respectively, for protein phosphorylation and ubiquitylation. It was unexpected that, however, both of the posttranscriptional modifications facilitated the stabilization of GW6a, which was different from the known mechanisms for the OsMAPK6-mediated control of grain size in rice. Meanwhile, our genetic and corresponding molecular evidence revealed that both of the viable *OsMAPK6-GW6a* and *CLG1-GW6a* axes play crucial roles in the regulation of rice grain size. In addition, we showed that the OsMAPK6-mediated phosphorylation and CLG1-mediated ubiquitylation of GW6a stabilize the substrate protein in a non-additive manner, and the triggered effects have non-additively altered grain growth in rice. Collectively, our findings revealed a previously unknown mechanism where the integration of several posttranscriptional modifications was utilized to fine-tune grain size in rice.

## Results

### Identification of mitogen-activated protein kinase OsMAPK6 and RING-type E3 ligase CLG1 as interacting partners of GW6a

We initially sought to identify additional interacting proteins of GW6a using a yeast two-hybrid (Y$_2$H) screening strategy. Ultimately, one of the putative GW6a interacting partners we identified was OsMAPK6, which was known as a positive modulator of rice grain size[8–10,15]. To verify the interaction, we resorted to pull-down assays. The *Escherichia coli*-produced fusion proteins GST-OsMAPK6 (or GST instead) and His-GW6a were incubated in a pull-down buffer along with anti-GST resin beads. Obviously, His-GW6a was pulled down by GST-OsMAPK6 (detected with an antibody to GW6a), but not by GST (Fig. 1a). We also confirmed the interaction using co-immunoprecipitation (Co-IP) assays in *Nicotiana benthamiana* (tobacco) leaves. Proteins were extracted from the leaves transiently co-expressing GW6a-Myc and OsMAPK6-GFP (or GFP instead); upon IP with an anti-GFP antibody, immunoblotting with anti-GFP and anti-Myc antibodies suggested that GW6a-Myc was immunoprecipitated by OsMAPK6-GFP, but not by GFP (Fig. 1b). Furthermore, we performed bimolecular flurescence complementation (BiFC) assays to verify the interaction, and observed strong fluorescence signals in the tobacco leaves co-expressing C-terminal part of yellow fluorescent protein fused with OsMAPK6 (OsMAPK6-cYFP) and N-terminal part of yellow fluorescent protein fused with GW6a (nYFP-GW6a), but did not in those leaves co-expressing OsMAPK6-cYFP and nYFP, or nYFP-GW6a and cYFP (Fig. 1c). Thus, these results suggest that OsMAPK6 interacts with GW6a.

In the meantime, we also paid special attention to another candidate GW6a interacting partner that is equivalent to the RING-type E3 ligase CLG1 (also called OsHRZ2)[22,23]. Similarly, we tested the interaction in a pull-down assay; as anticipated, CLG1-Myc was pulled down by GST-GW6a, but not by GST (Fig. 1d). The interaction was also verified using Co-IP assays, the results of which suggesting that GW6a-Myc was immunoprecipitated by CLG1-GFP, but not by GFP (Fig. 1e). Further BiFC assays revealed strong fluorescence signals in rice protoplasts co-expressing CLG1-cYFP and nYFP-GW6a, but did not in the corresponding controls (Fig. 1f). These results thus suggest that CLG1 is another interacting partner of GW6a. Collectively, we identified the mitogen-activated protein kinase OsMAPK6 and RING-type E3 ligase CLG1 that interact with GW6a.

### OsMAPK6 phosphorylates GW6a on Ser142 and Thr186 residues to control grain size

We inferred that GW6a might be a substrate of OsMAPK6. To test this idea, we performed kinase assays in tobacco leaves, and the following results indicated that co-expressing GFP fused with constitutively active version of OsMAPK6 (CA-OsMAPK6-GFP)[9] and GW6a-Myc could evidently increase the phosphorylation level of GW6a-Myc relative to that of the control (Fig. 2a). Consistent with the results, although in vitro kinase assays using a phos-tag SDS-PAGE gel of a low concentration (8%) with short exposure showed that MBP-GW6a cannot be visibly phosphorylated by GST tagged CA-OsMAPK6 (upper lane), the corresponding long exposure revealed that the phosphorylation seemingly occurred (lower lane), and MBP tagged N-terminal part of GW6a protein (amino acids 1-191; MBP-nGW6a) could be unambiguously phosphorylated independent of the addition of His-CA-OsMKK4, whereas MBP-cGW6a (amino acids 192-419) could not (Fig. 2b). In particular, in vitro kinase assays using a phos-tag SDS-PAGE gel of a high concentration (12%) showed that GW6a can be visibly phosphorylated by OsMAPK6, and calf intestinal alkaline phosphatase (CIAP) treatment could greatly compensate for the enhancement of GW6a phosphorylation (Fig. 2c). As we have anticipated, CA-OsMAPK6-GFP also enhanced the phosphorylation level of nGW6a-Myc in tobacco leaves (Fig. 2d). Accordingly, CIAP treatment could greatly compensate for the enhancement of His-nGW6a phosphorylation

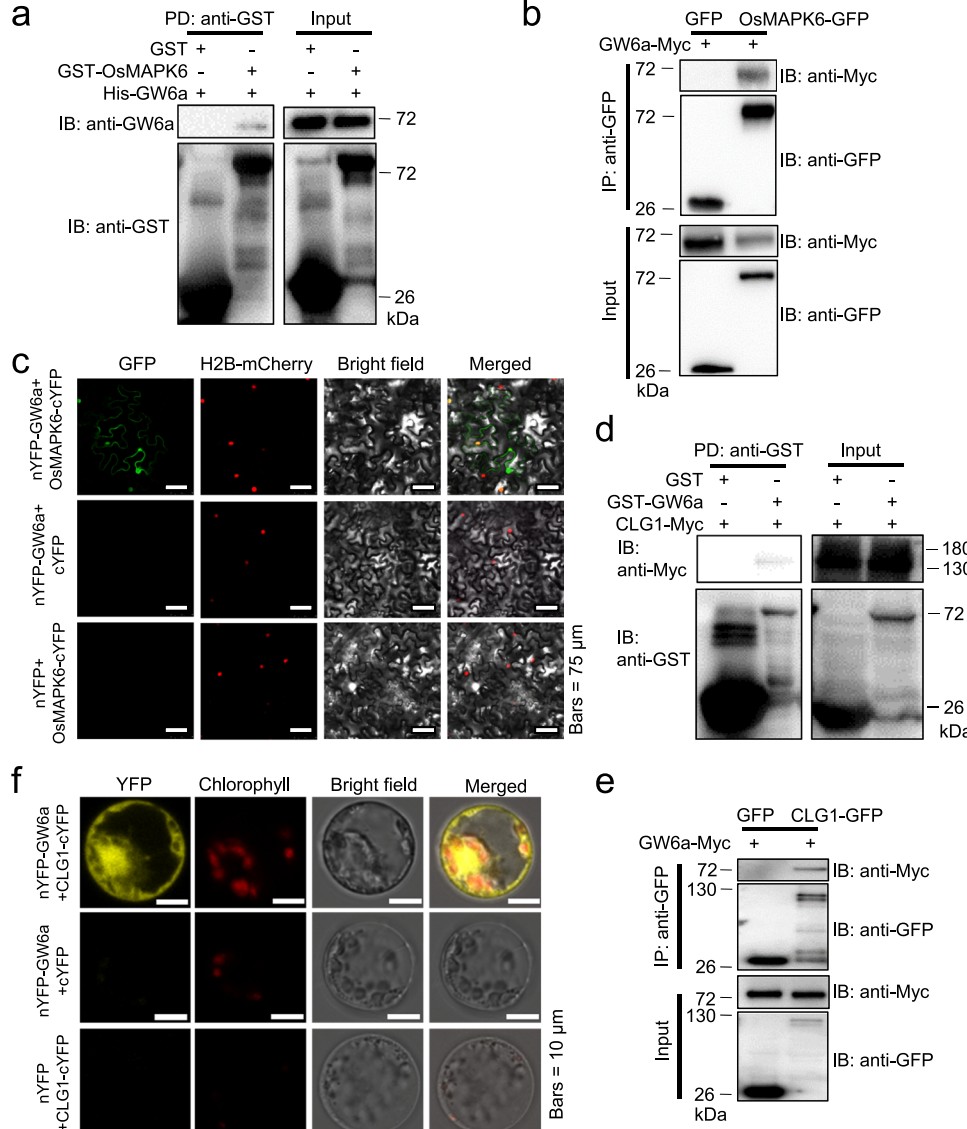

**Fig. 1 | Identification of OsMAPK6 and CLG1 that interact physically with GW6a.** **a** Pull-down assays showing that OsMAPK6 interacts with GW6a in vitro. PD pulling down, IB immunoblotting. **b** OsMAPK6-GFP interacts with GW6a-Myc in vivo. Total protein from *Nicotiana benthamiana* leaves co-expressing OsMAPK6-GFP (or GFP) and GW6a-Myc was immunoprecipitated by anti-GFP agarose beads and the immunoblots were probed using anti-Myc and anti-GFP antibodies. IP immunoprecipitation. **c** BiFC assays in *N. benthamiana* (tobacco) leaves of the interaction between nYFP-GW6a (nYFP) and OsMAPK6-cYFP (cYFP). The split YFP system was used. H2B-mCherry is a nucleus-localized marker. **d** Pull-down assays showing that CLG1 interacts with GW6a in vitro. **e** CLG1-GFP interacts with GW6a-Myc in tobacco leaves. **f** BiFC assays in rice protoplast cells of the interaction between nYFP-GW6a (nYFP) and CLG1-cYFP (cYFP). YFP fluorescence and autofluorescence signals from chloroplasts were pseudo-colored as yellow and red, respectively. The experiments in (**a**–**f**) were repeated at least two times will similar results.

caused by MBP-CA-OsMAPK6 (Fig. 2e). Hence, OsMAPK6 has the capability to phosphorylate GW6a.

Furthermore, we examined the GW6a phosphorylation sites targeted by OsMAPK6, and performed an in vitro kinase assay incubating His-nGW6a, GST-CA-OsMKK4, and MBP-CA-OsMAPK6; the reaction mixture was then subjected to liquid chromatography-tandem mass spectrometry. We thus identified two viable phosphorylation sites of GW6a (Ser142 and Thr186) located within the GNAT domain (Fig. 2f, Supplementary Fig. 1). To test the results, we mutated these amino acid residues to generate His tagged nGW6a$^{S142/T186D}$ (the two amino acid residues mutated into aspartic acid to mimic the phosphorylated nGW6a), nGW6a$^{S142D}$ and nGW6a$^{T186D}$, and examined their effects on GW6a phosphorylation utilizing in vitro kinase assays; the following results hinted that these mutations have distinctly altered the phosphorylation (Fig. 2g). Consistently, we found that the nGW6a phosphorylation levels of protein extracts of the tobacco leaves transiently

co-expressing CA-OsMAPK6-GFP and nGW6a$^{S142/T186D}$-Myc were markedly reduced compared with those co-expressing CA-OsMAPK6-GFP and nGW6a-Myc (Fig. 2h). In addition, we also mutated both of the amino acid residues into an alanine and co-expressed Myc tagged nGW6aS142/186A (or nGW6a instead) and CA-OsMAPK6-GFP in tobacco leaves, and found that OsMAPK6 also greatly losses the ability to phosphorylate the mutant version of nGW6a (Fig. 2i). These observations suggest that Ser142 and Thr186 of GW6a constitute the major sites phosphorylated by OsMAPK6.

These above results suggest that nGW6a, but not cGW6a, was markedly phosphorylated by OsMAPK6 (Fig. 2b, d, e), and coincidentally, we observed that relative to those of non-transgenic plants, the mature grains of transgenic rice plants over-expressing HA tagged nGW6a (*OE-HA-nGW6a*) and HA-cGW6a have increased grain length by over 14% and less than 5%, respectively (Supplementary Fig. 2). On the basis of these observations, we next

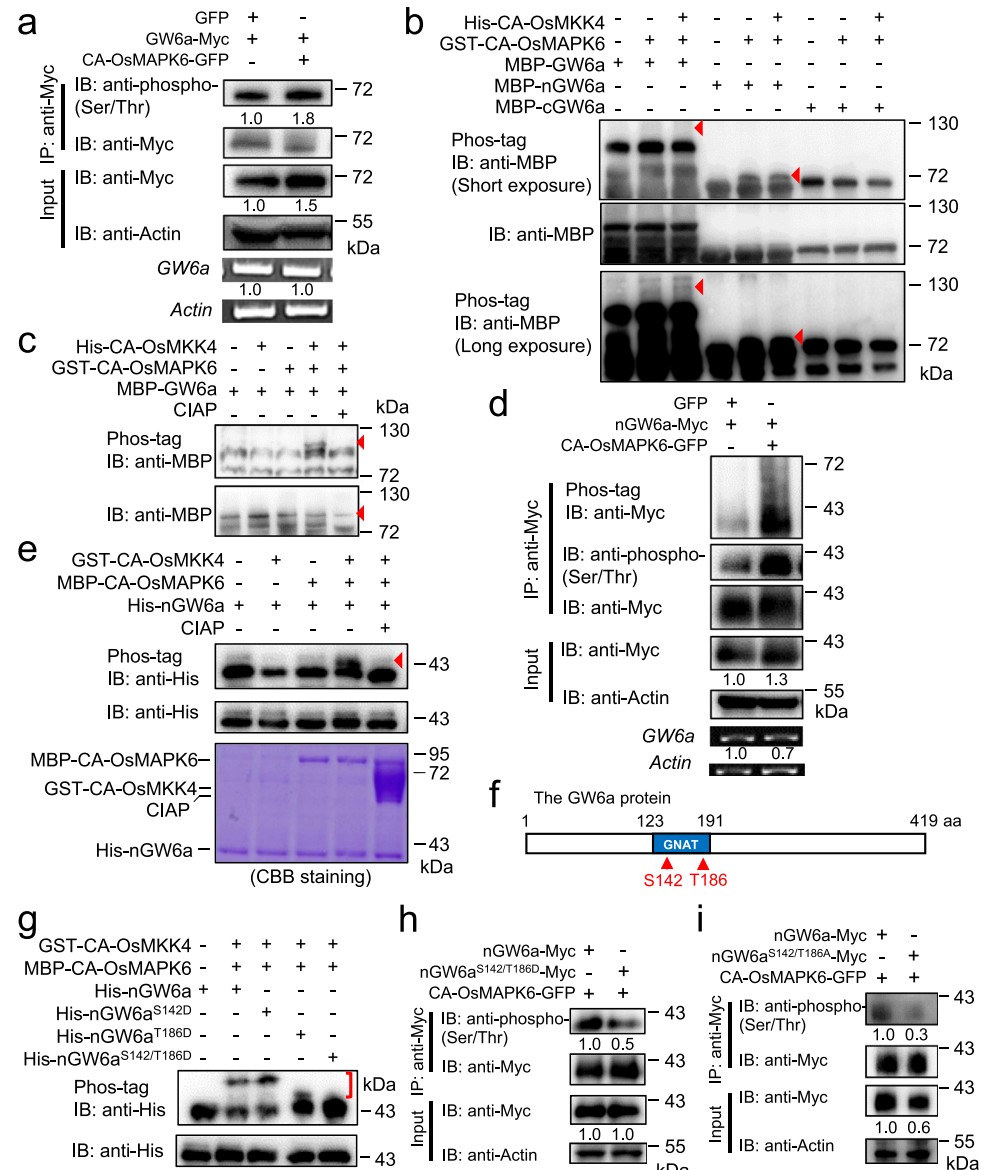

**Fig. 2 | GW6a is phosphorylated by OsMAPK6 at amino acid residues S142 and T186. a** Constitutively active version of OsMAPK6 (CA-OsMAPK6) phosphorylates GW6a. The CA-OsMAPK6-GFP (or GFP) and GW6a-Myc fusion proteins were expressed in tobacco leaves as indicated. IB immunoblotting. **b** GST-CA-OsMAPK6 phosphorylates MBP-nGW6a and possibly MBP-GW6a in vitro. The red arrowheads indicate phosphorylated MBP-nGW6a or phosphorylated MBP-GW6a. **c** GST-CA-OsMAPK6 phosphorylates MBP-GW6a in vitro, the biochemical effect of which was inhibited by calf intestinal alkaline phosphatase (CIAP) treatment for 1 h. The red arrowheads indicate MBP-GW6a (lower panel) or phosphorylated MBP-GW6a. **d** CA-OsMAPK6-GFP phosphorylates nGW6a-Myc in tobacco leaves.

**e** MBP-CA-OsMAPK6 phosphorylates His-nGW6a. The red arrowhead indicates phosphorylated His-nGW6a. **f** Schematic diagram showing the viable GW6a phosphorylation sites by Mass spectrometry. **g** MBP-CA-OsMAPK6 substantially loses the ability to phosphorylate His-nGW6a$^{S142/T186D}$ in vitro. The red bracket indicates shifted bands of phosphorylated His-nGW6a, His-nGW6a$^{S142D}$, His-nGW6a$^{T186D}$, and His-nGW6a$^{S142/T186D}$ in phos-tag SDS-PAGE gel. **h** CA-OsMAPK6-GFP markedly loses the ability to phosphorylate nGW6a$^{S142/T186D}$-Myc in tobacco leaves. **i** CA-OsMAPK6-GFP greatly loses the ability to phosphorylate nGW6a$^{S142/T186A}$-Myc in tobacco leaves. The experiments in (**a–e** and **g–i**) were repeated at least two times will similar results.

investigated whether the phosphorylation of GW6a is involved in grain size control, and produced transgenic rice lines overexpressing Myc tagged nGW6a$^{S142/T186D}$ (*OE-nGW6a$^{S142/T186D}$-Myc*) and nGW6a$^{S142/T186A}$ (the two amino acid residues mutated into alanine to mimic unphosphorylation; *OE-nGW6a$^{S142/T186A}$-Myc*). The *OE-nGW6a$^{S142/T186D}$-Myc* plants produced grains obviously larger than, whereas those of *OE-nGW6a$^{S142/T186A}$-Myc* set seeds almost comparable in size to did the control, although the two types of transgenic lines harbored similar transcriptional expression (Supplementary Fig. 3). These observations suggest that phosphorylation of GW6a by OsMAPK6 is involved in the regulation of grain size.

## GW6a is an ubiquitylation substrate of CLG1

We next investigated whether CLG1 was able to ubiquitinate GW6a. For this purpose, we obtained the fusion proteins His-GW6a and MBP-CLG1, and performed in vitro E3 ubiquitin ligase activity assays. In the presence of His-ubiquitin (His-Ub), His-E1 and E2, MBP-CLG1 can ubiquitinate His-GW6a (detected by anti-His and anti-Ub), whereas in the absence of any of His-E1, E2 or E3 enzymes, or His-Ub, we did not observe any His-GW6a ubiquitination signals (Fig. 3a). By contrast, MBP-CLG1△R (deletion of the RING domain of CLG1) was deprived of the ability to ubiquitinate GW6a (Fig. 3b). Consistent with these observations, the ubiquitylated GW6a-GFP levels of protein extract of

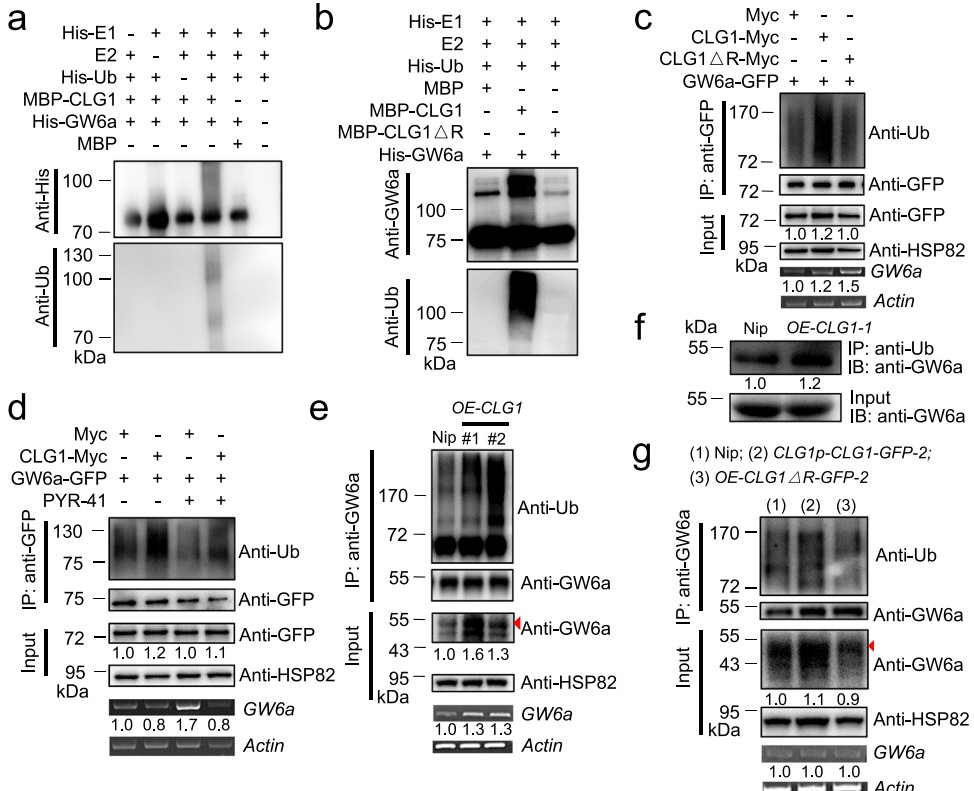

**Fig. 3 | CLG1 ubiquitylates GW6a in its RING domain-dependent manner. a** In vitro E3 ubiquitin ligase activity assays suggesting that CLG1 has the ability to ubiquitinate GW6a. MBP was used as a negative control. **b** CLG1ΔR (deletion of the RING domain of CLG1) loses the ability to ubiquitinate GW6a in vitro. **c** CLG1-Myc has, but CLG1ΔR-Myc loses the ability to ubiquitylate GW6a-GFP in rice protoplasts. **d** The ubiquitylation inhibitor PYR-41 inhibits the ubiquitination of GW6a-GFP by CLG1-Myc in rice protoplast cells. **e** The ubiquitylation levels of GW6a are enhanced in the transgenic young panicles of over-expression of *CLG1*. Total proteins from young panicles of *OE-CLG1* and the non-transgenic control ('Nip') were extracted, and immunoprecipitated with an antibody to GW6a. The ubiquitylation of GW6a levels are detected by following immunoblotting with anti-Ub and anti-GW6a antibodies. The arrowhead points to the GW6a band. **f** Enhanced ubiquitination of GW6a in *OE-CLG1* seedling leaves relative to that of the 'Nip' control. **g** The ubiquitylation levels of GW6a are enhanced in the young panicles of *CLG1p-CLG1-GFP*, but not in those of *OE-CLG1ΔR-GFP*. The arrowhead points to the GW6a band. The experiments in (**a–g**) were repeated at least two times will similar results.

rice protoplast cells co-expressing CLG1-Myc and GW6a-GFP were markedly enhanced, whereas those co-expressing CLG1ΔR-Myc and GW6a-GFP were not (Fig. 3c), echoing a conclusion that CLG1 ubiquitinates GW6a in a RING domain-dependent manner. The ubiquitylation inhibitor PYR-41 exerts an effect to block onset of ubiquitylation[28]. As expected, treatment of PYR-41 in rice protoplasts could largely inhibit the effect of CLG1 ubiquitylation of GW6a (Fig. 3d).

Supporting the above results, we monitored the ubiquitylation of GW6a in the protein extracts of *OE-CLG1* young panicles and seedling leaves, and the following results suggest that over-expression of *CLG1* clearly promoted the ubiquitylation levels of GW6a (Fig. 3e, f). Similarly, we found that *CLG1p-CLG1-GFP* also markedly enhanced the ubiquitylation of GW6a, whereas *OE-CLG1ΔR-GFP* did not (Fig. 3g). Thus, we concluded that GW6a is a viable ubiquitylation substrate of CLG1.

### Both modifications mediated by OsMAPK6 and CLG1 facilitate GW6a stabilization

We were curious to know the biochemical consequences of CLG1 and OsMAPK6-mediated modifications of GW6a. Interestingly, we noted that concomitant with the increased GW6a phosphorylation caused by OsMAPK6, GW6a stabilization was markedly enhanced (Fig. 2a, c). Coincidentally, along with the promoted GW6a ubiquitylation, its stabilization was also boosted by the CLG1 action as evidenced by some in vitro and in vivo assays (Fig. 3c–e, g), suggesting that both the OsMAPK6 and CLG1-mediated modifications of GW6a stabilize the substrate. To reinforce the conclusion, we evaluated the GW6a levels under the circumstance of co-expressing tag fused GW6a and CA-OsMAPK6 (or DN-OsMAPK6 instead, which changes the TEY activation loop motif of OsMAPK6 to AEF, causing loss of the catalytic activity)[9,29,30]. As expected, the GW6a levels were obviously enhanced by addition of the constitutive active version of CA-OsMAPK6, but not by that of DN-OsMAPK6 in both rice protoplast and tobacco systems (Fig. 4a, b). Supporting these observations, relative to that of ZH11, GW6a accumulation in the protein extracts of transgenic young panicles over-expressing CA-OsMAPK6 (*OE-CA-OsMAPK6*) was clearly boosted, but that of *OE-DN-OsMAPK6* was marginally increased (Fig. 4c). Considering that the mature transgenic rice grains of *OE-CA-OsMAPK6* became larger, whereas those of *OE-DN-OsMAPK6* were smaller[9,15], we inferred that OsMAPK6-mediated phosphorylation of GW6a is essential to stabilize the substrate to control grain size.

Similarly, we tested GW6a protein levels under the condition of co-expressing tag fused GW6a and CLG1 (or CLG1ΔR instead), and observed that GW6a abundance was significantly enhanced by addition of CLG1-GFP (or CLG1-Myc), but not by that of CLG1ΔR-GFP in both rice protoplast and tobacco systems (Fig. 4d, e). To support these results, we conducted in vivo degradation assays, and found that half-life ($T_{1/2}$) of GW6a-Myc in tobacco leaves co-expressing GW6a-Myc and CA-OsMAPK6-GFP was over two times that co-expressing GW6a-Myc and DN-OsMAPK6-GFP (Fig. 4f, g). Meanwhile, we found that $T_{1/2}$ of GW6a-Myc in tobacco leaves with addition of CLG1ΔR-GFP was almost half that with CLG1-GFP addition, which was comparable to that co-expressing GW6a-Myc and GFP (Fig. 4h, i). In addition, we investigated the changes of GW6a amount in cell components, and immunoblotting revealed that the altered GW6a-Myc levels arose predominantly in the nuclear fraction (Fig. 4j, k), which is consistent with a previous observation[26].

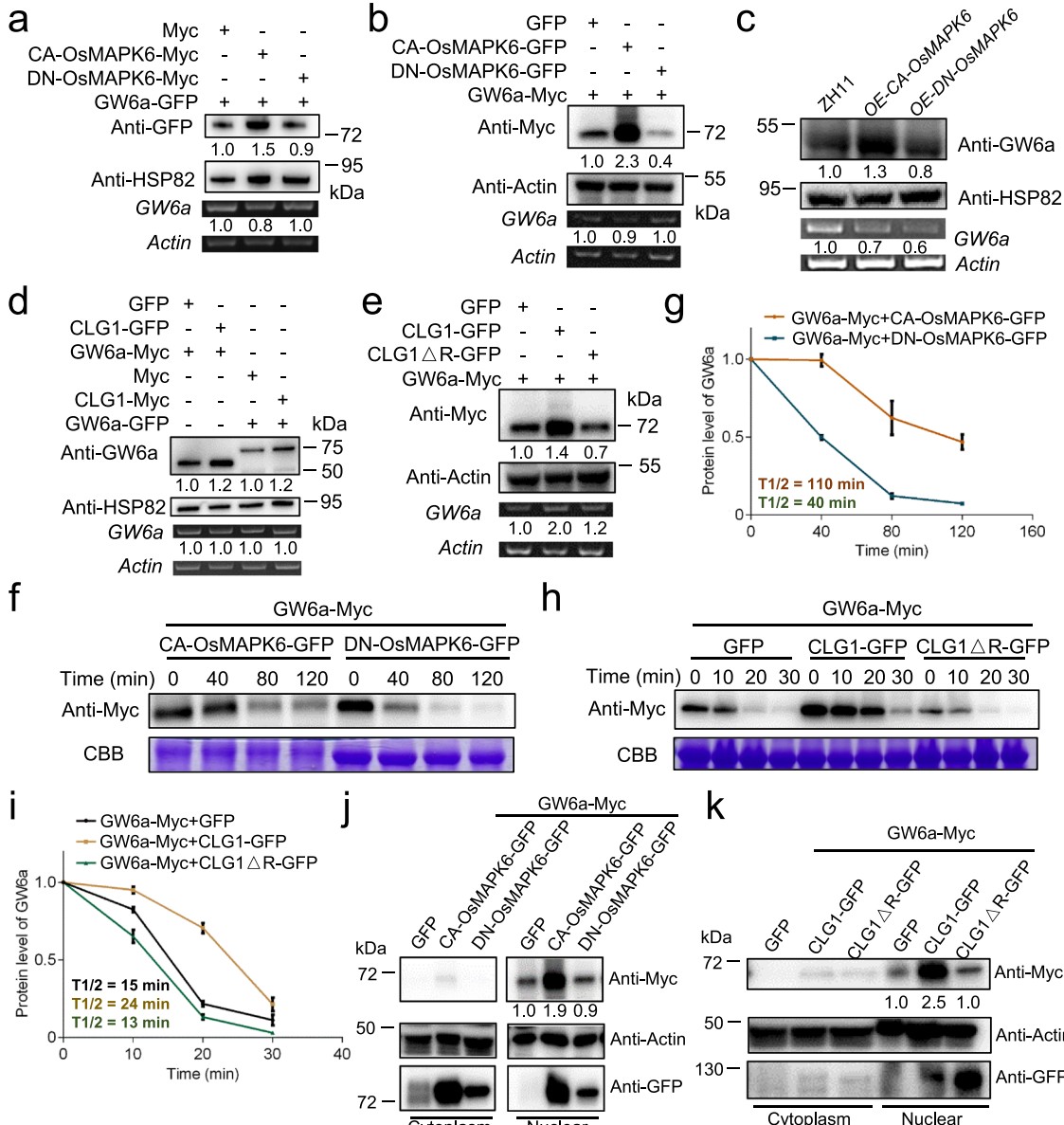

**Fig. 4 | Both posttranslational modifications by CLG1 and OsMAPK6 facilitate GW6a stabilization. a** CA-OsMAPK6-Myc, but not DN-OsMAPK6-Myc, substantially promotes the protein accumulation of GW6a-GFP in rice protoplasts. **b** CA-OsMAPK6-GFP, but not DN-OsMAPK6-GFP, substantially promotes the protein accumulation of GW6a-Myc in tobacco leaves. **c** The GW6a protein level is enhanced in the young panicles of *OE-CA-OsMAPK6*, but not in those of *OE-DN-OsMAPK6*. **d** The proteins levels of GW6a-Myc (or GW6a-GFP) are substantially enhanced with the addition of CLG1-GFP (CLG1-Myc) compared with those of addition of GFP (Myc) in rice protoplasts. **e** CLG1-GFP, but not CLG1ΔR-GFP, clearly promotes the protein accumulation of GW6a in transformed tobacco leaves. **f** In vivo degradation assays of GW6a-Myc by CA-OsMAPK6-GFP or DN-OsMAPK6-GFP in tobacco leaves. Protein extractions are incubated for the indicated time and used for immunoblotting assays. **g** A normalized plot for degradation of GW6a-Myc in

(**f**). Three independent biological repeats are performed for the analysis. The details of quantification and normalization are described under Methods. **h** In vivo degradation assays of GW6a-Myc by CLG1-GFP or CLG1ΔR-GFP, or GFP in tobacco leaves. Protein extractions are incubated for the indicated time and used for immunoblotting assays. **i** A normalized plot for degradation of GW6a-Myc in (**h**). Three independent biological repeats are performed for the analysis. **j** Immunobloting analysis of GW6a-Myc amount in the cytoplasm and nuclear components extracted from tobacco leaves co-expressing the indicated GW6a and CA-OsMAPK6 (or DN-OsMAPK6) fusion proteins. **k** Immunoblotting analysis of GW6a-Myc amount in the cytoplasm and nuclear components extracted from tobacco leaves co-expressing the indicated GW6a and CLG1 (or CLG1ΔR) fusion proteins. The experiments in (**a–f**, **h**, **j**, **k**) were repeated at least three times will similar results.

## OsMAPK6-GW6a and CLG1-GW6a define previously unknown genetic axes to control grain size through alteration of cell number

We next examined the genetic relationships between *OsMAPK6* and *GW6a* in grain size control. For this purpose, we crossed *OE-GW6a-Myc-1* with *OE-CA-OsMAPK6* to achieve *OE-CA-OsMAPK6/OE-GW6a-Myc-1* (Fig. 5a, b). The following phenotypic analyses showed that grain length and weight of *OE-CA-OsMAPK6/OE-GW6a-Myc-1* were

significantly longer and heavier than those of *OE-GW6a-Myc-1*, but shorter and lighter than those of *OE-CA-OsMAPK6* (Fig. 5c, d). In the meantime, we also obtained *OE-DN-OsMAPK6/OE-GW6a-Myc-2* by a genetic cross (Fig. 5e, f). As expected, we observed that grain length and weight of *OE-DN-OsMAPK6/OE-GW6a-Myc-2* was between those of *OE-GW6a-Myc-2* and *OE-DN-OsMAPK6* (Fig. 5g, h), hinting that *OE-GW6a-Myc* could suppress the inhibitory effect on grain size and weight by *Os-DN-OsMAPK6*. Thus, we conclude that *OsMAPK6* most

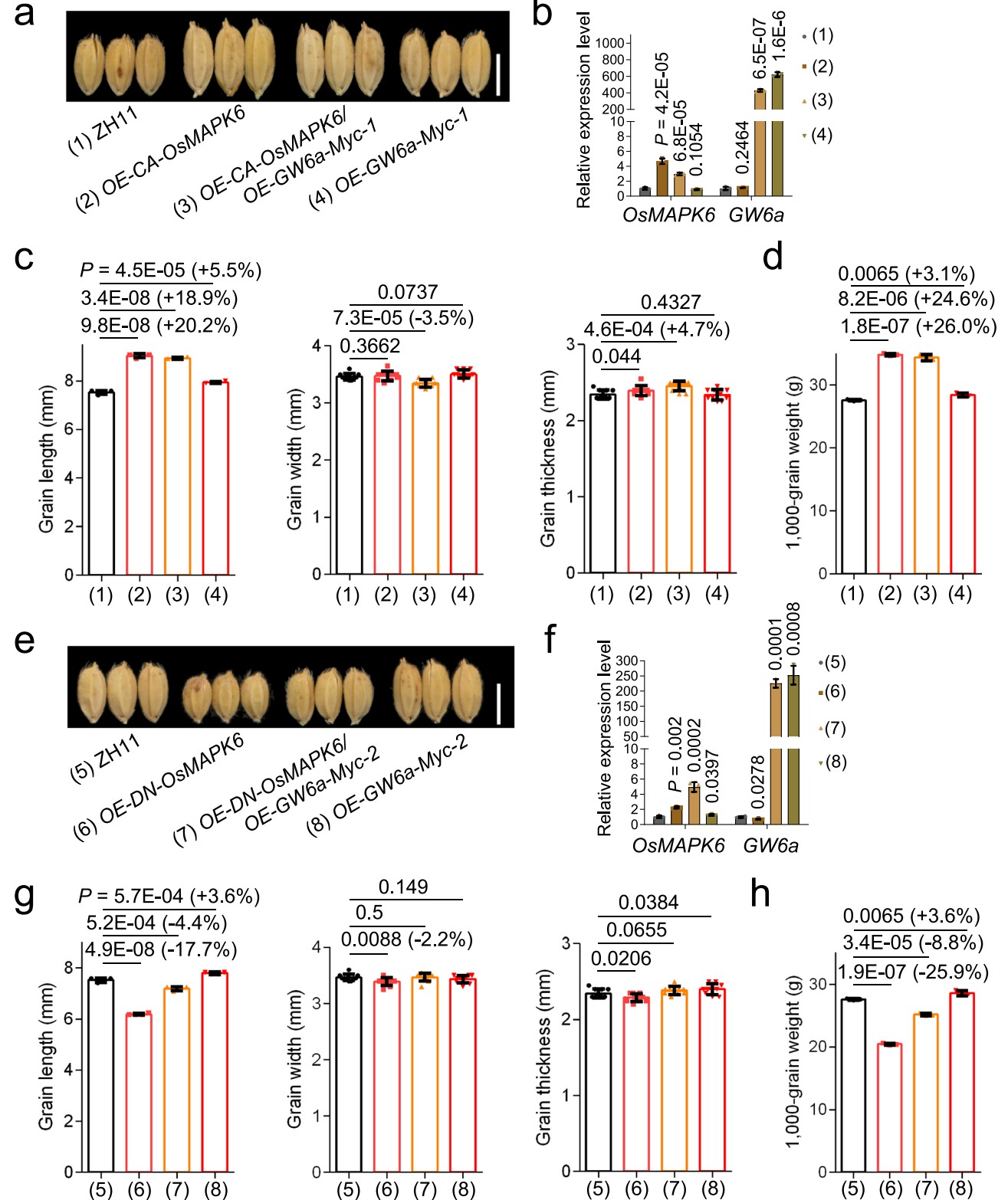

**Fig. 5 | *OsMAPK6* functions in a common genetic pathway with and acts upstream of *GW6a* to regulate grain length. a** Mature grains of 'ZH11', OE-CA-OsMAPK6 (pACTIN::CA-OsMAPK6), OE-CA-OsMAPK6/OE-GW6a-Myc-1, and OE-GW6a-Myc-1 (pSUPER::GW6a-Myc). Scale bar = 5 mm. **b** Relative expression of *OsMAPK6* and *GW6a* in plants shown in (**a**) (*n* = 3). The *UBQ5* was used as the internal reference gene. **c**, **d** Phenotypic comparisons of grain length (*n* = 4), grain width (*n* = 10), grain thickness (*n* = 10), and 1,000-grain weight (*n* = 3) in (**a**).

**e** Mature grains of 'ZH11', OE-DN-OsMAPK6 (pACTIN::DN-OsMAPK6), OE-DN-OsMAPK6/OE-GW6a-Myc-2, and OE-GW6a-Myc-2. Scale bar = 5 mm. **f** Relative expression of *OsMAPK6* and *GW6a* in plants shown in (**e**) (*n* = 3). The *UBQ5* was used as the internal reference gene. **g**, **h** Phenotypic comparisons of grain length (*n* = 4), grain width (*n* = 10), grain thickness (*n* = 10), and 1000-grain weight (*n* = 3) in (**e**). Data in (**b**–**d** and **f**–**h**) represent mean ± SD, and *P* values were obtained by one-way Student's *t*-test compared with the corresponding controls.

likely functions in the same genetic pathway as and acts upstream of *GW6a* to control grain size and weight.

CLG1 is a positive modulator of rice grain size (length), and transgenic rice with over-expression of a mutated form of CLG1 substituting three key amino acids in its RING domain set significant smaller grains in size (a dominant negative mutant)[23]. Consistent with these observations, our transgenic rice plants containing over-expression of *CLG1* or *CLG1* native promoter-driven CLG1-GFP fusion (*CLG1p-CLG1-GFP-1*) produced markedly increased grain length and weight (Supplementary Fig. 4a–d). In contrast, transgenic rice plants containing *OE-CLG1△R-GFP* set significantly shorter and lighter grains (Supplementary Fig. 4e–h). We then investigated the genetic relationship between *CLG1* and *GW6a* in the regulation of rice grain size, and generated the hybrid *OE-CLG1△R-GFP-1/OE-GW6a-Flag-1* through a genetic cross (Fig. 6a, b). As expected, we observed that the mature rice grains of *OE-CLG1△R-GFP-1/OE-GW6a-Flag-1* were significantly larger and heavier than those of *OE-CLG1△R-GFP-1* but smaller and lighter than those of *OE-GW6a-Flag-1* (Fig. 6c, d). We also found that the mature grains of the hybrid *CLG1p-CLG1-GFP-1/OE-GW6a-Flag-2* did not exhibit additive effects relative to those of *CLG1p-CLG1-GFP-1* and *OE-GW6a-Flag-2* (Fig. 6e–h). We also found that knockout of *GW6a* (*cg-gw6a*) significantly reduced grain length, although it profoundly harmed grain filling; obviously, the grain size (length) of *cr-gw6a/CLG1p-CLG1-GFP-1* was significantly longer than that of *cr-gw6a*, but slightly shorter than that of *CLG1p-CLG1-GFP-1* (Supplementary Fig. 5). Taken these results together, we concluded that *CLG1* and *GW6a* most likely function in the same genetic pathway to control grain size.

Previous studies revealed that both *OsMAPK6* and *GW6a* regulate grain length through change of cell number in spikelet hulls[8,27]. These observations prompted us to examine whether *CLG1* regulates grain length through alteration of cell number and/or cell size. To this end, we performed comparative analyses of the central parts of outer epidermal cells of mature grains between *OE-CLG1-1*, *CLG1p-CLG1-GFP-2*, and *OE-CLG1△R-GFP-2* and the corresponding control by scanning electron microscopy (Supplementary Fig. 6a, c). As expected, in the grain-length orientation, compared with those of non-transgenic control (Nip), the cell numbers of *OE-CLG1-1* and *CLG1p-CLG1-GFP-2* grains were significantly increased, with cell sizes unchanged (Supplementary Fig. 6b, d). By contrast, the cell numbers of *OE-CLG1△R-GFP-2* grains exhibited a significant decrease relative to those of the control (Supplementary Fig. 6e, f). These results suggest that like GW6a and OsMAPK6, CLG1 regulates grain size (length) through alteration of cell number in spikelet hulls.

### OsMAPK6 phosphorylation and CLG1 ubiquitylation of GW6a regulate grain size in a non-additive manner

To further test the relationship between *CLG1-GW6a* and *OsMAPK6-GW6a* axes in the regulation of grain size, we created hybrids between *OE-GW6a-Myc*, *OE-CA-OsMAPK6* and *OE-CLG1-GFP* and the triple transgenic lines (Fig. 7a). Phenotypic analyses revealed that compared with those of single and double over-expressions, the grain size and weight of the triple crossed lines did not exhibit additive effects (Fig. 7b, c), suggesting that the *CLG1-GW6a* and *OsMAPK6-GW6a* axes might correlate with each other rather than operate independently in the regulation of grain size. To support this assumption, we examined GW6a protein amounts in tobacco leaves transiently co-expressing a combination of GW6a-Myc and GFP alone or GFP fused with CLG1 or CA-OsMAPK6; as expected, the increase of GW6a amount did not exhibited additive effect with simultaneous CLG1-GFP and CA-OsMAPK6-GFP addition relative to CLG1-GFP or CA-OsMAPK6-GFP addition instead (Fig. 7d). We also obtained a very similar result in rice protoplast cells co-expressing the indicated fusion proteins (Fig. 7e). In addition, we evaluated GW6a protein levels in transgenic rice plants of the single, double, and the triple over-expression of *GW6a*, *CLG1*, and

*OsMAPK6*, and found that, as expected, the protein amount in the triple over-expression did not exhibit additive effect (Fig. 7f).

GW6a regulates grain size presumably through regulation of transcription[26,27]. We inferred that the *OsMAPK6-GW6a* and *CLG1-GW6a* effects could be seen in the transcriptional expression of their co-targeted genes. For this purpose, we compared the transcriptome of the transgenic *OE-GW6a-Myc*, *OE-CA-OsMAPK6*, and *CLG1p-CLG1-GFP* young panicles with that of the wild type. Ultimately, we identified 27 up-regulated genes shared by the three lines (Fig. 8a, b). It was worth noting that among these genes, *OsWRKY53* has been characterized as an important modulator of rice grain size through integration of BR signaling and MAPK pathway and *OsERF64* encodes an APETALA2/ethylene-response element binding protein[14,15]. Considering that APETALA2-type protein could play a key role in regulating rice grain size and yield[31], we selected *OsWRKY53* and *OsERF64* as target genes and validated by quantitative real-time PCR analysis that the transcriptional levels of the two genes were evidently up-regulated in these over-expression lines (Fig. 8c, d). We next examined the transcriptional levels of *OsWRKY53* and *OsERF64* in the young panicles of the single, double, and triple over-expression of *GW6a*, *CLG1*, and *OsMAPK6*, and unexpectedly, however, we found that the triple over-expression line did not exhibit additive effects on the expression of *OsERF64* but exhibited a different effect on that of *OsWRKY53* (Fig. 8e, f). In support of these observations, we transiently co-expressed a combination of tag fused GW6a, CA-OsMAPK6 and CLG1 proteins in rice protoplasts and examined the transcriptional levels of *OsERF64* and *OsWRKY53*; consistent with above observations, simultaneously expressing the three proteins did not exert additive effects on the expression of *OsERF64* but had a different influence on that of *OsWRKY53* (Fig. 8g, h).

## Discussion

In this current study, we have identified and defined the genetic regulatory axes *OsMAPK6-GW6a* and *CLG1-GW6a* that modulate grain size in a non-additive manner (Fig. 7). Mechanistically, the mitogen-activated protein kinase OsMAPK6 and E3 ligase CLG1 directly interacted with and targeted GW6a for phosphorylation and ubiquitylation, respectively and coincidentally, both of the modifications facilitated the stabilization of GW6a (Fig. 4). It was noteworthy that the OsMAPK6 phosphorylation of GW6a was involved in the regulation of grain size evidenced by the transgenic analyses in rice (Supplementary Fig. 3). In addition, consistent with the non-additive roles of the genetic axes in controlling grain size, the two posttranslational modifications exerted non-additive impact on the GW6a protein stability, and as well, on the transcriptional expression of their co-regulated downstream genes (Figs. 7, 8). Indeed, numerous proteins can be modified by multiple different types of posttranslational modifications whose effects on the biological functions of proteins could be synergistic or antagonistic, thus regulating distinct biological outcomes[32,33]. We have also performed pull-down assays to test whether phosphorylation of GW6a by OsMAPK6 affects its interaction with CLG1, by incubating the *E. coli*-produced GST fused GW6a, GW6a$^{S142/T186A}$ and GW6a$^{S142/T186D}$ with CLG1-Myc. Upon IP with GST, immunoblot assays using the anti-GST and anti-Myc antibodies suggested that compared with GST-GW6a, GST-GW6a$^{S142/T186D}$ exhibits much stronger binding to CLG1-Myc, whereas GST-GW6a$^{S142/T186A}$ has a much weaker binding (Supplementary Fig. 7), hinting that phosphorylation of GW6a by OsMAPK6 facilitates its interaction with CLG1, which presumably has a positive effect on the GW6a ubiquitylation by CLG1. Nevertheless, our findings have revealed correlations and interactions of multiple posttranslational modifications, i.e., protein phosphorylation, ubiquitylation and acetylation during rice grain development. However, to reveal the detailed mechanisms of how the viable interplay between OsMAPK6-mediated phosphorylation and CLG1-mediated ubiquitylation of GW6a operates to fine-tune grain size would be a considerable challenge, which awaits future experimentations.

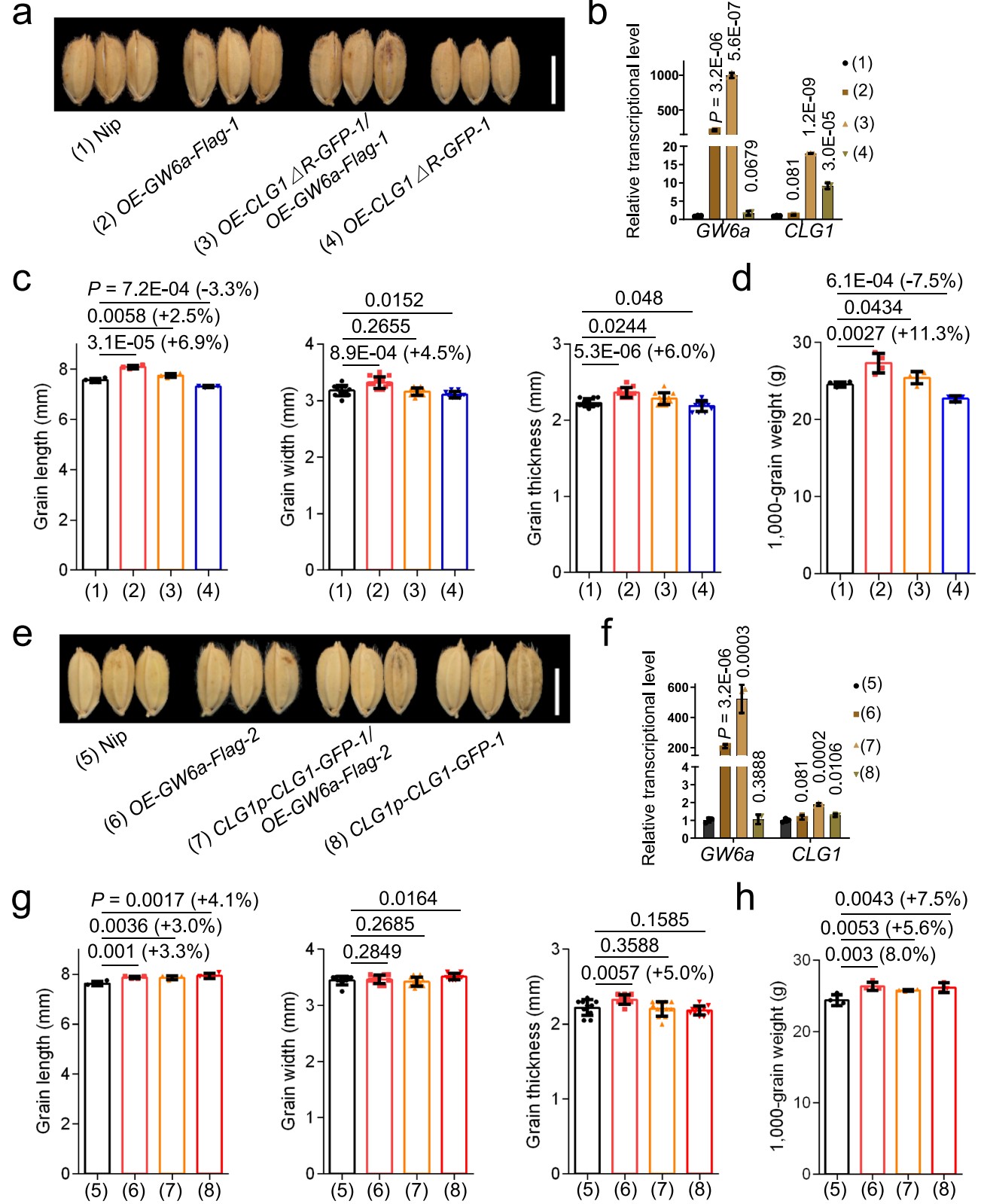

Posttranslational modifications, including protein phosphorylation and ubiquitination, play pivotal roles in regulating the stability, activity, subcellular localization, and interaction of the proteins being modified[33,34]. Especially, so far several substrates of OsMAPK6 have been identified in rice, which are intimately associated with the almighty functions of OsMAPK6 in plant growth and development. For instance, *OsWRKY53* positively regulates brassinosteroid signaling and grain size (length) in rice, the encoded protein of which was targeted by OsMAPK6 for phosphorylation that results in the enhancement of its transcriptional activity[14]. Similarly, OsMAPK6 also phosphorylates the finger transcription factor DST to enhance its transcriptional activation of *OsCKX2* to regulate spikelet number through modulation of cytokinin metabolism[35]. Furthermore, OsMAPK6 phosphorylates the raf-like kinase OsEDR1 at S861 that destabilizes OsEDR1 ultimately

**Fig. 6 | *CLG1* functions in the same genetic pathway as and acts upstream of *GW6a* to regulate grain size (length). a** Mature grains of Nip, *OE-GW6a-Flag-1* (*p35S::GW6a-Flag*), *OE-CLG1ΔR-GFP-1*, and OE-CLG1Δ*R-GFP-1/OE-GW6a-Flag-1*. Scale bar = 5 mm. **b** Relative transcriptional expression of *GW6a* and *CLG1* in Nip and the transgenic young panicles shown in (**a**). The *UBQ5* was used as the internal reference gene (*n* = 3). Comparisons of grain length (*n* = 4), grain width (*n* = 12), grain thickness (*n* = 12) (**c**), and 1,000-grain weight (*n* = 4) (**d**) between Nip and *OE-GW6a-Flag-1*, *OE-CLG1ΔR-GFP-1*, and OE-CLG1Δ*R-GFP-1/OE-GW6a-Flag-1*. Data in (**b**–**d**) represent mean ± SD, and *P* values were obtained by one-way Student's *t*-test compared with the corresponding controls. **e** Mature grains of Nip, OE-GW6a-Flag-2, CLG1p-CLG1-GFP-1, and CLG1p-CLG1-GFP-1/OE-GW6a-Flag-2. Scale bar = 5 mm. **f** Relative transcriptional expression of *GW6a* and *CLG1* between Nip and the transgenic young panicles shown in (**e**). The *UBQ5* was used as the internal reference gene (*n* = 3). Comparisons of grain length (*n* = 4), grain width (*n* = 10), grain thickness (*n* = 10) (**g**), and 1000-grain weight (*n* = 4) (**h**) between Nip and *OE-GW6a-Flag-2*, *CLG1p-CLG1-GFP-1*, and *CLG1p-CLG1-GFP-1/OE-GW6a-Flag-2*. Data in (**f**–**h**) represent mean ± SD, and *P* values were obtained by one-way Student's *t*-test compared with the corresponding controls.

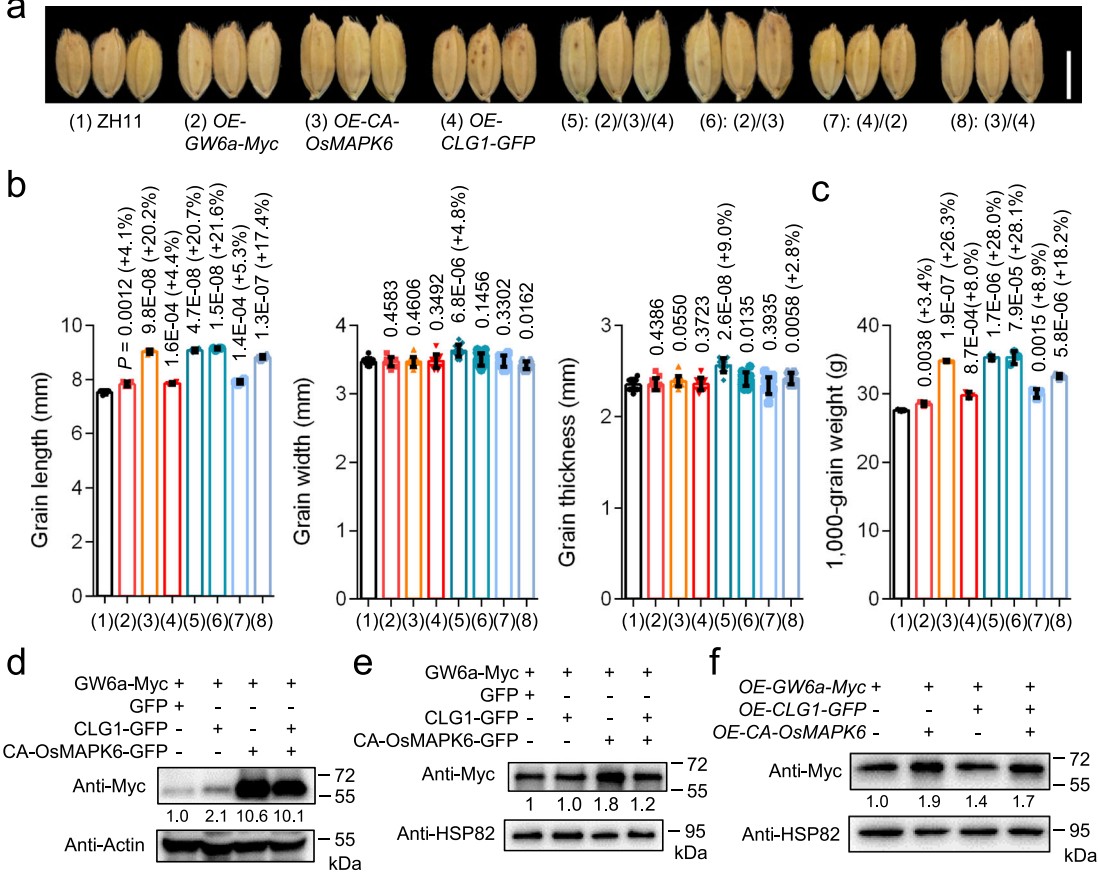

**Fig. 7 | Post-translation modifications mediated by OsMAPK6 and CLG1 stabilize GW6a in a non-additive manner to regulate grain size (length). a** Mature grains of ZH11, *OE-GW6a-Myc*, *OE-CA-OsMAPK6*, *OE-CLG1-GFP* (*pSUPER::CLG1-GFP*) and their hybrid progeny. Scale bar = 5 mm. Comparisons of grain length (*n* = 4), grain width (*n* = 12), grain thickness (*n* = 12) (**b**), and 1000-grain weight (*n* = 3) (**c**) between ZH11 and the indicated transgenic lines shown in (**a**). Data in (**b**, **c**) represent mean ± SD, and *P* values were obtained by one-way Student's *t*-test compared with the corresponding controls. **d** The protein levels of GW6a-Myc in tobacco leaves transiently co-expressing CLG1-GFP and/ or CA-OsMAPK6-GFP (or GFP instead). Image J software was used to quantify the protein levels of GW6a-Myc. **e** The protein levels of GW6a-Myc in rice protoplasts transiently co-expressing CLG1-GFP and/or CA-OsMAPK6-GFP (or GFP instead). Image J software was used to quantify the protein levels of GW6a-Myc. **f** The protein levels of GW6a in transgenic *OE-GW6a-Myc*, *OE-GW6a-Myc/OE-CA-OsMAPK6*, *OE-CLG1-GFP/OE-GW6a-Myc*, and *OE-GW6a-Myc/OE-CA-OsMAPK6/OE-CLG1-GFP* rice plants. Image J software was used to quantify the protein levels of GW6a-Myc. The experiments in (**d**–**f**) were repeated at least three times will similar results.

leading to enhanced resistance to rice disease[36]. In contrast, we showed that OsMAPK6 phosphorylates GW6a at S142 and T186 that facilitates the stability of GW6a to control grain size (Figs. 2, 4). Nevertheless, we have revealed a OsMAPK6-mediated mechanism for grain size control in rice.

In addition to the OsMAPK6-mediated phosphorylation, several GLYCOGEN SYNTHASE KINASE (GSK)-centered phosphorylation axes were described to play an important role in the regulation of rice grain size. In rice, the GSK3-like family has a total of 9 members that were classified into four major groups[37]. Of these members, OsGSK2, the central negative modulator of brassinosteroid signaling, interacts with and phosphorylates OVATE FAMILY PROTEIN 3 (OFP3) and MEI2-LIKE PROTEIN4 (OML4) to facilitate the protein stability of the modified substrates, whereas it phosphorylates and destabilizes DLT, OFP8, and SMOS1/RLA1[38–42]. OsGSK2 also interacts with the plant-specific transcription factor GROWTH REGULATING FACTOR 4 (OsGRF4) (also called GL2, GRAIN LENGTH ON CHROMOSOME 2) and inhibits its transcription activation activity to regulate grain size[43]. By contrast, another rice GSK3-like member OsGSK5 (also called TGW3) interacts with OsARF4 and facilitates OsARF4 accumulation, whereas it interacts with and phosphorylates OsIAA10 leading to OsIAA10 degradation by the 26 S proteasome pathway[44,45].

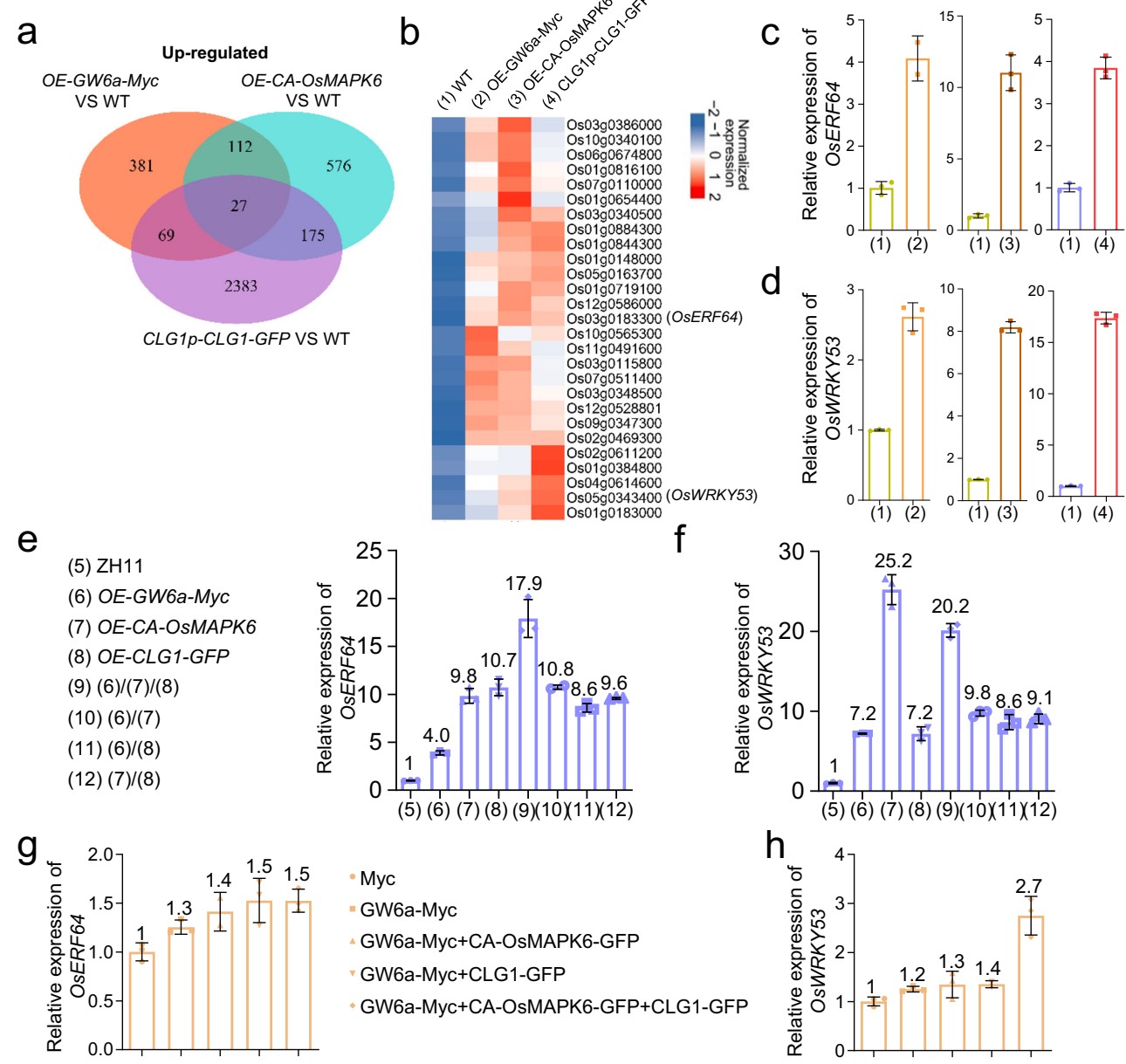

**Fig. 8 | *OsMAPK6-GW6a* and *CLG1-GW6a* exert non-additive effects on the transcriptional expression of their co-target genes. a** Venn diagrams showing the overlap of genes up-regulated in *OE-GW6a-Myc*, *OE-CA-OsMAPK6*, and *CLG1p-CLG1-GFP* compared to the corresponding WT as indicated. **b** Heatmap analysis of 27 up-regulated genes shared by *OE-GW6a-Myc*, *OE-CA-OsMAPK6*, and *CLG1p-CLG1-GFP* versus the corresponding WT. Relative transcriptional expression of *OsERF64* (**c**) and *OsWRKY53* (**d**) in WT, *OE-GW6a-Myc*, *OE-CA-OsMAPK6*, and *CLG1p-CLG1-GFP* young panicles as indicated in (**c**). Data are mean ± SD. Three biological replicates are performed for each assay. Relative transcriptional expression of *OsERF64* (**e**) and *OsWRKY53* (**f**) in young panicles of ZH11 (WT), and the single, double, and triple over-expression of *GW6a*, *OsMAPK6* and *CLG1* as indicated in (**e**). Data are mean ± SD. Three biological replicates are performed for each assay. Relative transcriptional expression of *OsERF64* (**g**) and *OsWRKYS3* (**h**) in rice protoplast cells expressing the Myc tag or a combination of fusion proteins as indicated in (**g**). Data are mean ± SD. Three biological replicates are performed for each assay.

Similarly, an early study showed that CLG1 ubiquitylates GS3 triggering its degradation via the endosome pathway[23]. In contrast, in this study, we observed that CLG1 targets GW6a for ubiquitination to stabilize the substrate (Fig. 4). Based on these above observations, we are surprisingly aware of an explicit fact that the action of these kinases and E3 ligases can steer the diverse substrates towards completely different fates through posttranslational modifications. We have revealed a non-proteolysis-mediated mechanism by which HDR3 increases the K63-linked poly-ubiquitination of and stabilizes GW6a to regulate grain size; by contrast, GW2 ubiquitylates and boosts the degradation of GW6a[26]. Similar to that of HDR3, the overexpression of

*CLG1* (*OE-CLG1-GFP*) has obviously increase of both K63-linked and K48-linked polyubiquitinations of GW6a (Supplementary Fig. 8a). By contrast, the overexpression of *GW2* (*OE-GW2*) has markedly and merely increased the modification of K48-linked polyubiquitinations of GW6a (Supplementary Fig. 8b). We reasoned that the different types of polyubiquitinations of GW6a by CLG1 and GW2 might at least partially contribute to altered GW6a fates of stability. Furthermore, to test whether monoubiquitylation is involved in the regulation of GW6a stabilization, we observed the GW6a ubiquitylation by utilizing the UbK0 variant (relative to the wild-type Ub), in which all seven lysine residues in Ub were changed into arginine to prevent the forming of

polyubiquitylation chains[28]. The following results suggest that both CLG1 and GW2 could monoubiquitylate GW6a in vitro. Moreover, it was worth noting that the three genetic factors *CLG1*, *HDR3*, and *GW6a* are positive regulators of grain length, but *GW2* negatively controls the trait, we reasoned that CLG1, HDR3, and GW2 could have a competitive interaction with GW6a. To test the assumption, we performed split firefly luciferase complementation assays of CLG1, HDR3, and GW2. We observed that the GW2 addition exerted a significantly negative effect on both the GW6a-CLG1 and GW6a-HDR3 interactions, while the individual addition of HDR3 and CLG1 did not significantly affect the GW2-GW6a interaction (Supplementary Fig. 9a–d). By contrast, the addition of HDR3 and CLG1, respectively, significantly enhanced the effect of the GW6a-CLG1 and GW6a-HDR3 interactions (Supplementary Fig. 9e, f). Thus, these primary results suggest that HDR3 and CLG1 synergistically enhance their interaction with GW6a, but GW2 behaves as an aggressive competitor of CLG1 and HDR3. However, to figure out the detailed mechanism underlying the discrepancy of biochemical consequences is a great challenge but deserves future investigations.

## Methods

### Plant materials and growth conditions

The rice (*Oryza sativa* L.) *japonica* cultivars 'Nipponbare' ('Nip') and 'Zhonghua 11' ('ZH11') were used for transgenic recipient as indicated. The transgenic over-expression lines of *OE-GW6a-Flag* were described previously[26]. The *ACTIN* promoter-*driven* transgenic over-expression lines of *OE-CA-OsMAPK6* and *OE-DN-OsMAPK6* were reported earlier[9]. To construct the 35 S *promoter*-driven transgenic *OE-CLG1* rice plants, the *CLG1* coding sequence (CDS) was inserted into *pCAMBIA1300*-GFP vector. To construct the *CLG1* promoter-driven transgenic *CLG1p-CLG1-GFP* rice plants, the *CLG1* CDS and 2-kb promoter were inserted into the *Bam*HI and *Spe*I sites of *pCAMBIA2300-eGFP* vector. To construct the *SUPER* promoter-driven transgenic *OE-CLG1△R-GFP* rice plants, *CLG1△R* (deletion of a segment spanning 2077-2208 bp of *CLG1* CDS) was inserted into the *pSuper1300-GFP* vector. To construct the *35 S promoter*-driven transgenic *OE-HA-nGW6a* and *OE-HA-cGW6*a rice plants, these truncated *GW6a* sequences were inserted into *pCAMBIA1300-GFP* vector. To produce the *OE-nGW6a-Myc, OE-nGW6a^{S142/T186D}-Myc*, and *OE-nGW6a^{S142/T186A}-Myc* rice plants, the wild-type and mutated *GW6a* CDS sequences were inserted into the *pSuper1300-Myc* vector for transgenic assays. To generate the *Super* promoter-driven transgenic *OE-GW6a-Myc* and *OE-CLG1-GFP* rice plants, the CDSs of *GW6a* or *CLG1* were inserted into the *pSuper1300-Myc* or *pSuper1300-GFP* vectors. To generate the *cr-gw6a* mutant, the single-guide RNAs (sgRNAs) was designed and the expression cassette was inserted into pBWA(V)H-Cas9 vector. The primers used in this work are listed in Supplementary Data 1. Transgenic rice plants were grown in experimental fields in Hainan (winter seasons) or Beijing (summer seasons). *Nicotiana benthamiana* (tobacco) plants were cultivated in growth chambers (16 h light/8 h dark at 25°C).

### Pull-down assays

The CDS of *GW6a* was individually cloned into the *Bam*HI and *Sal*I sites of pGEX-4T-1 and the *Bam*HI and *Hin*dIII sites of pET-32a to generate fusion proteins GST-GW6a and His-GW6a. The mutated GW6a CDS was cloned into the BamHI and SalI sites of the pGEX-4T-1 vector to express the fusion proteins GST-GW6a^{S142/T186D} and GST-GW6a^{S142/T186A}. The 4 × Myc and CDS of *CLG1* sequences were inserted into the multiple cloning site of pMAL-c2X to express a fusion CLG1-Myc protein. The CDS of *OsMAPK6* was cloned into the *Bam*HI and *Sal*I sites of pGEX-4T-1 to express GST-OsMAPK6. The constructs were transformed into *Escherichia coli* BL21 (DE3) and the recombinant proteins were induced by 0.5-mM IPTG (isopropyl-β-D-thio-galactopyranoside). For pull-down assays, GST-tag purification resin (Beyotime) were incubated with GST or GST-GW6a/GST-OsMAPK6 at 4°C for 1 h, and then

incubated with CLG1-Myc/His-GW6a at 4 °C for another 1 h. After incubation, the GST beads were washed thoroughly with PBS buffer (137 mM NaCl, 2.7 mM KCl, 10 mM Na$_2$HPO$_4$, 2 mM KH$_2$PO$_4$, 0.05% SDS, 1% Triton X-100), resolved by SDS-PAGE, and detected and probed with anti-GST (EASYBIO), or anti-Myc (TransGen), or anti-GW6a as indicated. Uncropped scans of immunoblotting results are shown in Source data.

### Co-IP assays

The CDS of *GW6a* was amplified and cloned into the *Hin*dIII and *Spe*I sites of the *pSuper1300-Myc* vector to express a fusion protein GW6a-Myc. The CDS of *CLG1* or *OsMAPK6* were cloned into *pSuper1300-GFP* to express CLG1-GFP or OsMAPK6-GFP. Each of the constructs was transformed into *Agrobacterium* EHA105 cells and co-infiltrated into tobacco leaves as indicated. After 72 h of cultivation, the leaves were harvested and ground in liquid nitrogen. Total proteins were extracted with a buffer (50 mM Tris-MES [PH 8.0], 10 mM EDTA [PH 8.0], 0.5 M sucrose, 1 mM MgCl$_2$, 1 mM DTT, 1 mM PMSF, 1 × complete protease inhibitor cocktail) and incubated with anti-GFP affinity beads (SMART) at 4°C for 3 h. The beads were washed three times with washing Buffer (10 mM Tris-HCl [PH 7.5], 0.5 mM EDTA [PH 8.0], 150 mM NaCl, 0.05% SDS, 1% Triton X-100, 1 mM DTT, 1 mM PMSF, 1 × complete protease inhibitor cocktail). Samples were resolved by SDS-PAGE, detected with anti-GFP (TransGen) and anti-Myc (TransGen). Uncropped scans of immunoblotting results are shown in Source data.

### BiFC assays

The CDS of *OsMAPK6* was cloned into pBI-2YC to generate a fusion protein OsMAPK6-cYFP. The CDS of *GW6a* was cloned into pBI-2YN to express nYFP-GW6a. Each construct was transformed into *Agrobacterium* EHA105 cells and co-infiltrated into tobacco leaves as indicated. Infiltration of EHA105 cells of H2B-mCherry into tobacco leaves serves as a localization signal for the nucleus. The fluorescence signals were examined using a confocal laser scanning microscope (Leica TCS SP5). The CDS of *CLG1* was cloned into the *Sal*I and *Sac*I sites of pSY735 to express CLG1-cYFP. The CDS of *GW6a* was cloned into the *Sal*I and *Bam*HI sites of pSY736 to express nYFP-GW6a. After co-transformation of the constructs, rice protoplasts were incubated overnight and YFP fluorescence signals were observed using a confocal laser scanning microscope (Leica TCS SP5). Proteins from rice protoplasts were extracted as described previously[26].

### Kinase assays in tobacco leaves

The wild-type (or variant) N-terminal (1–573 bp) and whole CDS of *GW6a* were cloned into *pSuper1300-Myc* to express nGW6a-Myc, nGW6a^{S142/T186D}-Myc, nGW6a^{S142/T186A}-Myc, and GW6a-Myc. *CA-OsMAPK6* was cloned into the *Hin*dIII and *Spe*I sites of *pSuper1300-GFP* to express CA-OsMAPK6-GFP. Each of these constructs was transformed into *Agrobacterium* EHA105 cells and co-infiltrated into tobacco leaves as indicated. Total leave protein was extracted with buffer (50 mM Tris-MES [PH 8.0], 10 mM EDTA [PH 8.0], 0.5 M sucrose, 1 mM MgCl$_2$, 1 mM DTT, 1 mM PMSF, 1 × complete protease inhibitor cocktail) and incubated with anti-Myc Nanobody Agarose Beads (KT HEALTH) at 4 °C for 3 h. The beads were washed for three times with washing buffer (10 mM Tris-HCl [PH 7.5], 0.5 mM EDTA [PH 8.0], 150 mM NaCl, 1 mM DTT, 1 mM PMSF, 1 × complete protease inhibitor cocktail). The samples were analyzed by SDS-PAGE with 50 μM Phos-tag and detected with anti-phospho-(Ser/Thr) (Abcam) or anti-Myc. Uncropped scans of immunoblotting results are shown in Source data.

### In vitro kinase assays and phosphopeptide analysis

*CA-OsMKK4* was cloned into pGEX-4T-1 or pET-32a to express GST-CA-OsMKK4 or His-CA-OsMKK4, respectively. *CA-OsMAPK6* was cloned into pGEX-4T-1 or pMAL-c2X to express GST-CA-OsMAPK6 or MBP-CA-OsMAPK6, respectively. The *nGW6a^{S142/T186D}*, *nGW6a^{S142D}*, and

$nGW6a^{T186D}$ sequences were amplified with the indicated primers and cloned into pEASY-blunt3 (TransGen, CB301) for sequencing. The cDNA segments of $nGW6a$, $nGW6a^{S142/T186D}$, $nGW6a^{S142D}$, and $nGW6a^{T186D}$ and $cGW6a$ were then cloned into pET-32a or pMAL-c2X to express His-nGW6a, MBP-nGW6a, MBP-cGW6a, respectively. For an in vitro kinase assay, the indicated purified proteins of 1 μg His-CA-OsMKK4 (or GST-CA-OsMKK4), 1.5 μg GST-CA-OsMAPK6 (or MBP-CA-OsMAPK6), and 1.5 μg MBP-nGW6a (or His-nGW6a, His-nGW6a$^{S142D}$, His-nGW6a$^{T186D}$, His-nGW6a$^{S142/T186D}$) and 1.5 μg MBP-cGW6a were incubated in reaction buffer (25 mM Tris-HCl [PH 7.5], 10 mM MgCl$_2$, 1 mM DTT, 200 μM ATP) at 30 °C for 1 h. The samples were then analyzed by 8% or 12% SDS-PAGE added with 50 μM Phos-tag (Wako, Phos-tag Acrylamide AAL-107) and detected by immunoblotting with anti-His or anti-MBP antibodies. After in vitro kinase assays, proteins were digested with trypsin (Promega). The digested peptide mixtures were injected into a mass spectrometer (ORBITRAP FUSION LUMOS) for Mass spectrometry analysis. The phosphorylated residues in MBP-nGW6a were found by using Proteome Discoverer software. Uncropped scans of immunoblotting results are shown in Source data.

### E3 ubiquitin ligase activity assays

The in vitro E3 ubiquitin ligase activity assay was performed referring to a previous description with some modifications[46]. The coding sequences of $CLG1$ and $CLG1\triangle R$ were cloned into pMAL-c2X to express MBP-CLG1 and MBP-CLG1$\triangle$R, respectively. $GW6a$ was cloned into pET-32a to express His-GW6a. The CDS of $GW2$ was cloned into pET-32a to express His-GW2. Then, 1 μg E3s (His-GW2, MBP-CLG1 or MBP-CLG1$\triangle$R), 2 μg His-GW6a, 100 ng His-E1 (R&D Systems), 250 ng E2 (R&D Systems), and 5 μg His-Ub (R&D Systems) or His-UbK0 (all seven lysine residues in ubiquitin were changed to arginine, R&D Systems) were incubated in a 30-μL reaction buffer (50 mM Tris-HCl [pH 7.5], 10 mM MgCl$_2$, 2 mM DTT, and 5 mM ATP) at 30 °C for 2 h. Polyubiquitinated proteins were detected by immunoblotting with anti-Ub, anti-MBP, anti-His, and anti-GW6a antibodies.

To examine CLG1 ubiquitylation of GW6a in rice protoplasts, $pSuper1300$-$GW6a$-$GFP$ (expressing GW6a-GFP) was co-transformed with $pSuper1300$-$CLG1\triangle R$-$Myc$ (expressing CLG1△R-Myc) or $pSuper1300$-$CLG1$-$Myc$ (expressing CLG1-Myc) into rice protoplasts. Total proteins were extracted and immunoprecipitated with anti-GFP affinity beads (a final concentration of 50 μM PYR-41 (Sigma) was applied as shown). The beads were then washed 3 times with washing buffer (10 mM Tris-HCl [PH 7.5], 0.5 mM EDTA [PH 8.0], 150 mM NaCl, 1 mM DTT, 1 mM PMSF, 1 x complete protease inhibitor cocktail). Ubiquitylation was detected by immunoblotting with anti-Ub.

To analyze the CLG1 or GW2 ubiquitination of GW6a in transgenic samples, total proteins from 1 g of seedling leaves or young panicles of $OE$-$CLG1$, $CLG1p$-$CLG1$-$GFP$, $OE$-$CLG1\triangle R$-$GFP$, and $OE$-$GW2$ were extracted for immunoprecipitation assays as indicated. The rProtein A/G magnetic beads (SMART) of 50 μL conjugated with 10 μg of anti-Ub or anti-GW6a antibodies were incubated with the total protein extracts at 4 °C for 3 h. The ubiquitination levels of GW6a were detected by immunoblotting with anti-Ub, anti-UbK48 (CST), anti-UbK63 (CST), or anti-GW6a antibodies. Uncropped scans of immunoblotting results are shown in Source data.

### Protein degradation assay

The constructs pSuper1300-CA-OsMAPK6, pSuper1300-DN-OsMAPK6-GFP, pSuper1300-GW6a-Myc, pSuper1300-GFP, pSuper1300-CLG1-GFP, and pSuper1300-CLG1△R-GFP were individually transfected into Agrobacterium EHA105 cells, and co-infiltrated into tobacco leaves as indicated. Total leaves proteins were extracted with degradation buffer (25 mM Tris-HCl [PH 7.5], 10 mM NaCl, 10 mM MgCl$_2$, 5 mM DTT, 4 mM PMSF, and 2 mM ATP) at room temperature. Total GW6a-Myc proteins were then analyzed by SDS-PAGE and detected by immunoblotting with anti-Myc. Coomassie Brilliant Blue

staining was used as protein loading control. Three independent biological repeats were performed for the analysis of quantification and normalization of the GW6a protein levels. Uncropped scans of immunoblotting results are shown in Source data.

### Nuclear/cytoplasmic fractionation and quantification of proteins

The constructs $pSuper1300$-$GW6a$-$Myc$, $pSuper1300$-$CA$-$OsMAPK6$-$GFP$, $pSuper1300$-$DN$-$OsMAPK6$-$GFP$, $pSuper1300$-$CLG1$-$GFP$, $pSuper1300$-$GFP$, and $pSuper1300$-$CLG1\triangle R$-$GFP$ were co-transformed into tobacco leaves as indicated. Total proteins from transformed leaves were extracted by grinding on rice with Honda buffer (2.5% Ficoll 400, 5% Dextron T40, 0.4 M Sucrose, 25 mM Tris-HCl [PH 7.4], 100 mM MgCl$_2$, 5 mM PMSF, 5 mM DTT, 1 × complete protease inhibitor cocktail) and centrifuged at 1,500 g for 20 min to obtain crude nuclear and cytoplasmic fractions. The supernatant containing cytoplasmic fraction was centrifuged at 16,000 g for 15 min to remove the possible cellular debris. The above pellet containing nuclear fraction was dissolved in ChIP extraction buffer (0.25 M Sucrose, 10 mM Tris-HCl [PH 8.0], 10 mM MgCl$_2$, 1% Triton X-100, 0.1 mM PMSF, 1 mM DTT, 1 x complete protease inhibitor cocktail) and centrifuged at 16,000 g for 10 min at 4 °C. Ultimately, the pellet was re-suspended in nuclear lysis buffer (50 mM Tris-MES [PH 8.0], 10 mM EDTA [PH 8.0], 0.5 M sucrose, 1 mM MgCl$_2$, 1 mM DTT, 1 mM PMSF, 1 x complete protease inhibitor cocktail). The samples were analyzed by SDS-PAGE and detected by immunoblotting with anti-Myc and anti-Actin antibodies. Protein levels were quantified using Image J software. Image J software was used for the quantitative analysis of the relative protein abundance of the WB results[47]. Briefly, for example, in Fig. 4A of our manuscript, changing the format and removing possible background interference of the protein bands of interest was performed in the software. Furthermore, the relative density data for the control (the sample extracted from the rice protoplast cells co-expressing Myc-tag and GW6a-GFP) to the HSP82 (or Actin instead in other Figures) protein integrated density was defined as being equal to 1.0, and the relative density data of another sample (for example, the sample extracted from the rice protoplasts co-expressing CA-OsMAPK6-Myc and GW6a-GFP) divided by the relative control value led to the indicated value (1.5). Similarly, we have obtained the other relative sample values in this study. Uncropped scans of immunoblotting results are shown in Source data.

### Split firefly luciferase complementation assay

The CDS of GW6a, CLG1, GW2, and HDR3 were individually cloned into pCAMBIA1300-nLUC or pCAMBIA1300-cLUC to express GW6a-nLUC, CLG1-cLUC, GW2-cLUC, and HDR3-cLUC. The CDS of CLG1, GW2, and HDR3 were cloned into pSuper1300-GFP, respectively, to express CLG1-GFP, GW2-GFP, and HDR3-GFP in tobacco leaves. Each construct was transformed into the *Agrobacterium* EHA105 cells and co-infiltrated into tobacco leaves as indicated. Tobacco leaves were submerged in a solution with luciferin (YEASEN) for 48 h to capture LUC signals using a CCD imaging system (Tanon).

### RNA extraction and quantitative real-time PCR analysis

Total RNA was extracted using the RNA Kit (ZOMANBIO) and reverse transcription was performed using a kit (TIANGEN FastQuart). *ACTIN1* was used as an internal reference gene to normalize the expression of GW6a in RT-PCR. Real-time quantitative PCR was performed on a Bio-Rad CFX96 system. Rice *UBQ5* was used to normalize the data and relative expression levels were calculated using $2^{-\triangle\triangle Ct}$ method. The primers used in this work are listed in Supplementary Data 1.

### Histological analysis

For cytological analysis of cell size and number of spikelet hulls, the central parts of lemmas of mature seeds of *OE-CLG1*,

CLG1p-CLG1-GFP, OE-CLG1△R-GFP and the corresponding non-transgenic control were treated and observed with a scanning electron microscope (Hitachi). The longitudinal cell length was quantified using Image J software. Cell number was decided by grain length and the average cell length.

### RNA sequencing and analysis

Young panicles (5 cm in length) of OE-GW6a-Myc, OE-CA-OsMAPK6, CLG1p-CLG1-GFP, and the corresponding wild type were collected and immediately frozen in liquid nitrogen. The RNA sequencing was performed using Illumina NovaSeq 6000. Raw data were trimmed and then mapped to the Nipponbare genome database. Differential expression of genes was analyzed using the DESeq2 program and the genes were considered to be differentially expressed when the FDR was less than 0.05 and the relative fold was higher than 1.5.

### Statistics & reproducibility

Data are shown as mean ± SD (standard deviation), calculated using GraphPad Prism 8.0 version. Statistical analysis and number of biologically independent sample (n) were indicated in the figure legends. Significant differences were determined with one-way Student's *t*-test by using Microsoft Excel 2021 software.

### Reporting summary

Further information on research design is available in the Nature Portfolio Reporting Summary linked to this article.

## Data availability

All materials in this study are available from the corresponding authors upon request. The authors declare that all data supporting the findings of this study are available within the article, supplementary information files, and source data. The primers used in this study are provided as Supplementary Data 1. The RNA-seq data generated in this study has been deposited in NCBI SRA database under accession number PRJNA979276. Sequence data from this study can be found in the GenBank database under the following accession numbers: GW6a (Os06g0650300), CLG1 (Os05g0551000), OsMAPK6 (Os06g0154500), OsMKK4 (Os02g0787300), OsWRKY53 (Os05g0343400), OsERF64 (Os03g0183300), UBQ5 (Os05g0160200), GW2 (Os02g0244100), OsHDR3 (Os03g0267800), ACTIN1 (Os03g0718100). Source data are provided with this paper.

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

## Acknowledgements

We would like to thank Prof. Lei Wang (Institute of Botany) for sharing the plasmid vectors pBI-2YN and pBI-2YC, and Prof. Yuxin Hu (Institute of Botany) for sharing the plasmid vectors *pSuper1300-GFP* and *pSuper1300-Myc*. We also thank Dr. Xiuping Xu, Dr. Zhuang Lu, and Dr. Jingquan Li (Institute of Botany) for technical assistance. This work was supported by STI 2030-Major Projects (2023ZD0406902), the Strategic Priority Research Program of Chinese Academy of Sciences (XDA24010101-2), the National Key Research and Development Program of China (2016YFD0100402), and the National Natural Science Foundation of China (91735302, 91435113, and 31471466).

## Author contributions

X.-J.S. designed the study. C.B., G.-J.W., X.-H.F., and Q.G. performed most of the experiments and analyzed the data. W.-Q.W., R.X., S.-J.G., S.-Y.S., M.M., W.-H.L., C.-M.L, and Y.-H.L. conducted some of the experiments. C.B. and X.-J.S. wrote the manuscript.

## Competing interests

The authors declare no competing interests.
