## [Peer Review File · Nature Communications]

REVIEWER COMMENTS

Reviewer #1 (Remarks to the Author):

In this manuscript, Bai et al. reported that the grain size regulator GW6a is subject to the OsMAPK6-mediated phosphorylation and CLG1-mediated ubiquitylation, which non-additively stabilizes GW6a to control rice grain size. The authors specified the major phosphosites on GW6a and revealed that the phosphorylation and ubiquitylation both enhance GW6a stability, in contrast to most reported cases in protein ubiquitylation. The work presented an interesting story that two PTMs synergistically promote the function of their substrate to control grain size. A clear logic flow, supported by biochemical and genetic evidence, leads to well-presented conclusions. My concerns are as follows.

1. Figure 2 needs improvement and more detailed narration, as I am pretty confused by the unspecific bands in Fig 2B, I could not tell the phosphorylation status of MBP-6a, though the authors concluded that MBP-tagged GW6a and cGW6a were not phosphorylated. If so, how could the truncated nGW6a be phosphorylated, but the full version protein could not? Does this mean that the OsMAPK6-mediated phosphorylation actually does not work in rice plants since GW6a, instead of nGW6a, is the natural, functional version of rice? Or does a nGW6a protein exist in rice plants to work on it?
2. Fig2F, a routine way to determine the phosphosites of a protein is to mutate the site into an Alanine, mimicking the blocked site. Is it a typo here that the sites were mutated into D, or should the authors provide a more detailed description of the experiment design? Even though, S142D had almost identical phospho-intensity as the nGW6a, meaning this site is not, or at least not, a major phosphosite by MAPK6.
3. Figure 2G, the band intensity should be quantified as those in 2A.
4. Figure S2, I'm curious about the grain size of OE-HA-GW6a since the N and C truncated proteins have been presented here.
5. L233-239, the logic is not proper. The higher amount of GW6A-Myc in Figure 2A doesn't necessarily mean that the phosphorylation stabilizes GW6A; it could be the differences in co-transformation efficiency, mRNA transcription, or quite a lot other reasons. I would suggest the deletion of this description.
6. Similarly, for Figure 4A-D, the mRNA transcription level of all the samples should be provided to confirm that the variation in protein abundance is ascribed to the protein stability, not or partially because of mRNA transcription.
7. In figure 4A-C, it is weird to see that the addition of DN-OsMAPK6 could increase (4C), decrease (4B), and keep (4A) the protein abundance of GW6a in three different systems.

Reviewer #2 (Remarks to the Author):

GW6a is a previously cloned gene with a positive effect on rice grain size. This study identified two GW6a-interacting proteins, OsMAPK6 and CLG1, with known roles in grain size regulation. OsMAPK6 can phosphorylate GW6a and CLG1 mediates the ubiquitylation of GW6a. Interestingly, both post-translational modifications stabilize GW6a. Further genetic and molecular results provide more clues to reveal their roles in coordinating grain size regulation. Although the study provides extensive data on the post-translational regulation of the GW6a protein, the biological significance of this regulation is still unclear and several issues remain to be addressed.

1. In most cases, ubiquitination will destabilize the target protein. However, CLG1-mediated ubiquitination of GW6a stabilizes GW6a. Surprisingly, in another paper published by the same group (Gao et al., *Plant Cell*, 2021: 33: 3331-3347), GW2 can also ubiquitylate GW6a. However, the effect of GW2 promoted the degradation of GW6a. Why?
2. Since GW2, HDR3, and CLG2 are all involved in the ubiquitylation and stability of GW6a, please interpret how they coordinate the regulation of rice grain size by modulating GW6a.
3. Why can only the N-terminal part of the GW6a protein be phosphorylated by MAPK6, while the full-length GW6a and the C-terminal part of GW6a cannot? What's the biological significance?
4. Authors should provide a clear and detailed description of how to quantitatively evaluate the protein abundance of the WB results. For example, the quantification data of the WB bands in Fig 4A-E, J and K.
5. It's strange that the change of GW6a protein abundance in Fig. 4B and 4C was opposite. Specifically, in the presence of DN-OsMAOK6, the protein abundance of GW6a was decreased in Fig. 4B but increased in Fig. 4C.
6. The difference in protein abundance between different lanes was subtle in some figures, such as Fig. 3C, Fig. 7E and Fig. 7F. I suggest the authors to confirm the results with more biological replicates.
7. In Fig. 4H, why was the protein abundance of GW6a-Myc in the first lane (0 min) of each group so different?
8. The authors mentioned that "grain length and weight of OE-DN-OsMAPK6/OE-GW6a-Myc-2 did not exhibit additive genetic effects relative to those of OE-GW6a-Myc-2 and OE-DN-OsMAPK6, which were very similar to those of OE-GW6a-Myc-2 (Figures 5G, 5H)". The description is inaccurate and inconsistent with the Fig. 5G and 5H.
9. Please explain why you chose two different systems for BiFC analysis. One in tobacco leaves (Figure 1C) and one in rice protoplast cells (Fig. 1F).

Reviewer #3 (Remarks to the Author):

Seed size and shape is important agronomic traits which determine the yield and quality in rice. GW6a, MAPK6, and CLG1 are reported as important seed size regulators via diverse mechanisms. However, the regulatory relationship among these three components is not known. In this manuscript, author identified two interaction partner of GW6a, a kinase MAPK6 and a E3 ligase CLG1. Then they demonstrate that MAPK6 and CLG can phosphorylate and ubiquitinate GW6a, respectively. Additionally, they mapped the phosphorylation site of GW6a by MAPK6. Interestingly, they found both MAPK6 and CLG promote the stabilization of GW6a. further, they show GW6a function downstream of MAPK6 and CLG via analyzing the seed size phenotype of double mutant. Finally, they identified several common regulated genes of GW6a, MAPK6, and CLG1 through RNA-Seq. Collectively, they suggest that two kinds of posttranslational modification of GW6a by MAPK6 and CLG1 might represent a novel seed size regulatory mechanism. The result is interesting, nevertheless, there have some points need to be clarified.

1. In this manus, author try to reveal the underlying regulatory mechanism among GW6a, MAPK6, and CLG1. However, they are three well-known seed shape regulator, which affect novelty of study. In addition, the intensity and depth of study is not enough, and some underlying deep-seated mechanism is not investigated and revealed, like whether phosphorylation of GW6a by MAPK6 affect ubiquitination of GW6a?
2. Some results seems contradictory. They show overexpression of both NGW6a and CGW6a can increase seed size, implying that both NGW6a and CGW6a are functional. How NGW6a and CGW6a perform their biological function? Why overexpression of NGW6a and CGW6a show similar increased seed size phenotype? In addition, NGW6a, but not CGW6a, can be phosphorylated by MAPK6. These results might imply that phosphorylation of GW6a is not necessary for its biological function.
3. It is known that mono-ubiquitination and poly-ubiquitination modification promote the stability and degradation of target protein, respectively. In this manus, CLG1 promote the stabilization of GW6a, but in most of ubiquitination assay in this study, it looks like the GW6a is poly-ubiquitinated by CLG1. So the ubiquitination of GW6a by CLG1 is mono-ubiquitination or poly-ubiquitination, which need to be further investigated.
4. In most the double mutant analysis, the overexpression of GW6a, CLG, and MAPK6 plants was used. As ectopic expression of gene will result in many unnatural phenotype, which affect the observed conclusion. For genetic analysis, the knockout mutant will be better than transgenic plants with ectopic expression.

Title: "OsMAPK6 phosphorylation and CLG1 ubiquitylation of GW6a non-additively enhance rice grain size through stabilization of the substrate"

Tracking Number: NCOMMS-23-49960

Authors: Bai et al.

Reviewer Comments:

Reviewer #1: In this manuscript, Bai et al. reported that the grain size regulator GW6a is subject to the OsMAPK6-mediated phosphorylation and CLG1-mediated ubiquitylation, which non-additively stabilizes GW6a to control rice grain size. The authors specified the major phosphosites on GW6a and revealed that the phosphorylation and ubiquitylation both enhance GW6a stability, in contrast to most reported cases in protein ubiquitylation. The work presented an interesting story that two PTMs synergistically promote the function of their substrate to control grain size. A clear logic flow, supported by biochemical and genetic evidence, leads to well-presented conclusions.

Response: We thank this reviewer expert for your positive comments.

Comments: My concerns are as follows. 1. Figure 2 needs improvement and more detailed narration, as I am pretty confused by the unspecific bands in Fig 2B, I could not tell the phosphorylation status of MBP-6a, though the authors concluded that MBP-tagged GW6a and cGW6a were not phosphorylated. If so, how could the truncated nGW6a be phosphorylated, but the full version protein could not? Does this mean that the OsMAPK6-mediated phosphorylation actually does not work in rice plants since GW6a, instead of nGW6a, is the natural, functional version of rice? Or does a nGW6a protein exist in rice plants to work on it?

Response: We thank the reviewer expert for the insightful comments!

Our experimental results of kinase assays in tobacco leaves suggest that the

constitutively active version of OsMAPK6 can markedly phosphorylate GW6a (Figure 1A). Furthermore, although *in vitro* kinase assays using a phos-tag SDS-PAGE gel of a low concentration (8%) with short exposure showed that the full-length GW6a could not be visibly phosphorylated by OsMAPK6 (upper lane), the corresponding long exposure revealed that the phosphorylation seemingly occurs (lower lane of Figure 1B). In particular, our recent *in vitro* kinase assays using a phos-tag SDS-PAGE gel of a high concentration (12%) showed that GW6a could be visibly phosphorylated by OsMAPK6, and CIAP treatment could greatly compensate for the enhancement of GW6a phosphorylation (Figure 1C). Collectively, we concluded that GW6a could be phosphorylated by OsMAPK6 in both *in vivo* and *in vitro* conditions. We revised our manuscript accordingly.

Fig.1 The full-length GW6a can be phosphorylated by OsMAPK6.

Comments: 2. Fig2F, a routine way to determine the phosphosites of a protein is to mutate the site into an Alanine, mimicking the blocked site. Is it a typo here that the sites were mutated into D, or should the authors provide a more detailed description of the experiment design? Even though, S142D had almost identical phospho-intensity as the nGW6a, meaning this site is not, or at least not, a major phosphosite by MAPK6.

Response: We thank the reviewer expert for the critical comments.

Fig. 2 The *E. coli*-produced His-fused nGW6a^{S142/T186A}, nGW6a^{S142A}, and nGW6a^{T186A} are always concomitant with unspecific protein bands.

We agree with the reviewer expert that mutating an amino acid residue into alanine (A) that mimics the blocked site is the most commonly used method to determine the phosphosite of a protein. However, it is unfortunately that the *E. coli*-produced His-fused nGW6a^{S142A}, nGW6a^{T186A}, and nGW6a^{S142/T186A} are always concomitant with unspecific protein bands (Figure 2A), which would inevitably cause trouble in the judgment of effects of phosphorylation by OsMAPK6. Thus, considering that to mutate the amino acid residue into aspartic acid (D) in essence could also effectively prevent the site from further direct phosphorylation by the kinase protein, we generated the S142/T186-to-D mutations and examined the biochemical effect on GW6a phosphorylation by OsMAPK6. Nevertheless, our experimental results from *in vitro* assays and assays in tobacco leaves suggest that the amino acid residues S142 and T186 of GW6a constitute the major sites phosphorylated by OsMAPK6 (see Figure 2 of our manuscript). To reinforce the conclusion, we also mutated both of these amino acid residues into an alanine and co-expressed Myc tagged nGW6a^{S142/186A} (or nGW6a instead) and CA-OsMAPK6-GFP in tobacco leaves, and found that OsMAPK6 also greatly losses the ability to phosphorylate the mutant version of nGW6a (Figure 2B).

We revised our manuscript accordingly.

Comments: 3. Figure 2G, the band intensity should be quantified as those in 2A.

Response: Thank you very much for the comments.

Following the suggestions by the reviewer expert, we have quantified the band intensity in Figure 2G of our manuscript (Figure 3).

Fig. 3 CA-OsMAPK6-GFP markedly loses the ability to phosphorylate nGW6a^{S142/186D}-Myc in tobacco leaves.

Comments: 4. Figure S2, I'm curious about the grain size of OE-HA-GW6a since the N and C truncated proteins have been presented here.

Response: We thank the reviewer expert for the comments.

We assumed that the reviewer expert wanted to know the comparisons of the grain size related traits between the transgenic rice over-expression of GW6a, nGW6a, cGW6a, and the corresponding control. Coincidentally, we have cultivated the transgenic rice with constructs of *OE-GW6a-Flag*, *OE-HA-nGW6a*, and *OE-HA-cGW6a* in the same paddy field in the summer of 2022. Obviously, among these transgenes, the mature rice grains of *OE-HA-nGW6a* were the longest (largest), although the exogenous transcription levels of which were apparently lower than those of *OE-GW6a-Flag* and *OE-HA-cGW6a* in the young panicles (Figures 4A, and 4B). Phenotypic measurement and statistical analysis showed that relative to

the grain length of WT, those of *OE-GW6a-Flag*, *OE-HA-nGW6a*, and *OE-HA-cGW6a*, respectively, has a significant increase by 4.1-4.7%, 14.8-16.5%, and 5.5-5.9%, respectively (Figure 4C). As expected, 1,000-grain weight of the mature grains of *OE-HA-nGW6a* was the heaviest among these transgenes (Figure 4D). We revised our manuscript accordingly.

Fig. 4 *OE-HA-nGW6a* has much longer and heavier grains than that of *OE-HA-cGW6a* and *OE-GW6a-Flag*.

Comments: 5. L233-239, the logic is not proper. The higher amount of GW6A-Myc in Figure 2A doesn't necessarily mean that the phosphorylation stabilizes GW6A; it could be the differences in co-transformation efficiency, mRNA transcription, or quite a lot other reasons. I would suggest the deletion of this description.

Response: We thank the reviewer expert for your insightful comments.

Our logic arrangement of the referred descriptions (Lines 233-239) aimed to

provide an important clue as to the biochemical consequence of OsMAPK6 and CLG1-mediated modifications of GW6a. These primary observations revealed a positive correlation between the level of GW6a phosphorylation by OsMAPK6 (and also of its ubiquitylation by CLG1) and the GW6a protein accumulation (Figure 5), and therefore promoted us to conducted a series of experiments to test a conclusion that these modifications facilitate GW6a stabilization. Thus, although the higher amount of GW6A-Myc and the concomitant increased GW6a phosphorylation don't necessarily mean that the phosphorylation stabilizes GW6A, we assumed this positive correlation is indispensable that points to a viable conclusion. Following the suggestions by the reviewer expert, we have added the *GW6a* and *Actin* mRNA transcription results to our manuscript (Figure 5).

Fig. 5 There is a positive correlation between the level of GW6a phosphorylation by

OsMAPK6 (and of its ubiquitylation by CLG1) and the GW6a protein accumulation.

Comments: 6. Similarly, for Figure 4A-D, the mRNA transcription level of all the samples should be provided to confirm that the variation in protein abundance is ascribed to the protein stability, not or partially because of mRNA transcription.

Response: We thank this reviewer expert for the comments.

Following the suggestions by this and other reviewer experts, we have performed the corresponding experiments for additionally several times and provided the mRNA transcription levels of *GW6a* and *Actin* (as a control) of all the samples in Figures 4A-4E of our manuscript (Figures 6 and 7). The resultant observations suggest that the mRNA transcription levels of *GW6a* did not show any obvious and consistent correlation with those of *GW6a* protein abundance (Figure 6). We revised our manuscript accordingly.

Fig. 6 Both modifications by CLG1 and OsMAPK6 increase GW6a stability.

Comments: 7. In figure 4A-C, it is weird to see that the addition of DN-OsMAPK6 could increase (4C), decrease (4B), and keep (4A) the protein

abundance of GW6a in three different systems.

Response: We thank the reviewer expert for your constructive critiques.

Fig. 7 Results of several repeated experiments confirm that the modification by OsMAPK6 facilitate GW6a stabilization.

To solve the confusion that the addition of DN-OsMAPK6 exerted a different

effect on the protein abundance of GW6a in the three different systems, we have recently conducted the referred experiments with discretion for additionally 4-5 times (Figure 7). We thus ultimately chose the representative results for each of these systems (indicated by the red checkmark; Figures 6 and 7). Based on these observations, we favor a conclusion that the DN-OsMAPK6 addition had a negative effect on the protein abundance of GW6a in different systems. We revised our manuscript accordingly.

Reviewer #2: GW6a is a previously cloned gene with a positive effect on rice grain size. This study identified two GW6a-interacting proteins, OsMAPK6 and CLG1, with known roles in grain size regulation. OsMAPK6 can phosphorylate GW6a and CLG1 mediates the ubiquitylation of GW6a. Interestingly, both post-translational modifications stabilize GW6a. Further genetic and molecular results provide more clues to reveal their roles in coordinating grain size regulation. Although the study provides extensive data on the post-translational regulation of the GW6a protein, the biological significance of this regulation is still unclear and several issues remain to be addressed.

Response: The positive comments by this reviewer expert are very much appreciated.

Comments: 1. In most cases, ubiquitination will destabilize the target protein. However, CLG1-mediated ubiquitination of GW6a stabilizes GW6a. Surprisingly, in another paper published by the same group (Gao et al., Plant Cell, 2021: 33: 3331-3347), GW2 can also ubiquitylate GW6a. However, the effect of GW2 promoted the degradation of GW6a. Why?

Response: We thank the reviewer expert for the critical comments.

We have revealed the non-proteolysis-mediated mechanism by which HDR3 interacts with and increases the K63-linked poly-ubiquitination of GW6a and stabilizes GW6a to regulate grain size; by contrast, GW2 also interacts with,

ubiquitylates and boosts the degradation of GW6a (Gao et al., 2021). Similar to that of HDR3, the overexpression of *CLG1* (*OE-CLG1-GFP*) has obviously increase of both K63-linked and K48-linked polyubiquitinations of GW6a (Figure 8A). By contrast, the overexpression of *GW2* (*OE-GW2*) has markedly and merely increased modification of the K48-linked polyubiquitinations of GW6a (Figure 8B). We reasoned that the different types of polyubiquitination of GW6a by CLG1 and GW2 might at least partially contribute to altered GW6a fates of stability. We added these descriptions to our manuscript.

Fig. 8 CLG1 and GW2 ubiquitylate GW6a in a different manner.

Comments: 2. Since GW2, HDR3, and CLG2 are all involved in the ubiquitylation and stability of GW6a, please interpret how they coordinate the regulation of rice grain size by modulating GW6a.

Response: We thank the reviewer expert for the critical comments.

Fig. 9 GW2 has a stronger ability than HDR3 and CLG1 to interact with GW6a.

Frankly speaking, the topic of the mechanism how GW2, HDR3, and CLG1 coordinate the regulation of rice grain size by modulating GW6a has gone somewhat beyond of the scope of this study. We have reported that HDR3 interacts with and stabilizes GW6a to regulate rice grain size, and GW2 also interacts with and ubiquitylates but boosts the degradation of GW6a (Gao et al., 2021). On the basis of these results and the fact that the three genetic factors *CLG1*, *HDR3*, and *GW6a* are positive regulators of grain length, but *GW2* negatively controls the trait, we tentatively assumed that *CLG1*, *HDR3*, and *GW2* could have a competitive interaction with *GW6a*. To test the assumption, we have recently performed split firefly luciferase complementation assays of *CLG1*, *HDR3*, and *GW2*. We observed that the *GW2* addition exerted a significantly negative effect on both the *GW6a-CLG1* and *GW6a-HDR3* interactions, while the individual addition of *HDR3* and *CLG1* did not significantly affect the *GW2-GW6a* interaction (Figures 9A-9D). By contrast, the addition of *HDR3* and *CLG1*, respectively, significantly enhanced the effect of the *GW6a-CLG1* and *GW6a-HDR3* interactions (Figures 9E, and 9F). Thus, these primary results suggest that *HDR3* and *CLG1* synergistically enhance their interaction with *GW6a*, but *GW2* behaves as the aggressive competitor of

CLG1 and HDR3. We added the results to our manuscript.

Comments: 3. Why can only the N-terminal part of the GW6a protein be phosphorylated by MAPK6, while the full-length GW6a and the C-terminal part of GW6a cannot? What's the biological significance?

Response: Thank you very much for the insightful comments.

We realized that it is so brusque to conclude that the full-length GW6a cannot be phosphorylated by OsMAPK6. It is worth noting that our experimental results of kinase assays in tobacco leaves suggest that the constitutively active version of OsMAPK6 can markedly phosphorylate GW6a (Figure 1A).

Furthermore, although *in vitro* kinase assays using a phos-tag SDS-PAGE gel of a low concentration (8%) with short exposure did not suggest the full-length GW6a as a direct target phosphorylated by OsMAPK6 (upper lane), the long exposure revealed that OsMAPK6 phosphorylation of GW6a most likely occurs (lower lane of Figure 1B). In particular, our recent *in vitro* kinase assays using a phos-tag SDS-PAGE gel of a high concentration (12%) showed that GW6a could be visibly phosphorylated by OsMAPK6, and CIAP treatment could greatly compensate for the enhancement of GW6a phosphorylation (Figure 1C). Thus, we concluded that the full-length GW6a (and C-terminal part of GW6a) could be phosphorylated by OsMAPK6 in both *in vivo* and *in vitro* conditions. We revised our manuscript accordingly.

Comments: 4. Authors should provide a clear and detailed description of how to quantitatively evaluate the protein abundance of the WB results. For example, the quantification data of the WB bands in Fig 4A-E, J and K.

Response: Thank you for the helpful comments.

Image J software was used for the quantitative analysis of the relative protein abundance of the WB results (Schneider et al., 2012). Briefly, for example, in Figure 4A of our manuscript, changing the format and removing possible

background interference of the protein bands of interest was first performed in the software. Furthermore, the relative density data for the control (the sample extracted from the rice protoplast cells co-expressing Myc-tag and GW6a-GFP) to the HSP82 (or Actin instead in other Figures) protein integrated density was defined as being equal to 1.0, and the relative density data of another sample (for example, the sample extracted from the rice protoplasts co-expressing CA-OsMAPK6-Myc and GW6a-GFP) divided by the relative control value led to the indicated value (1.5). Similarly, we have obtained the other relative sample values in this study, for example, in Figures 4B-4E, 4J and 4K. We added these descriptions to the method of our manuscript.

Comments: 5. It's strange that the change of GW6a protein abundance in Fig. 4B and 4C was opposite. Specifically, in the presence of DN-OsMAOK6, the protein abundance of GW6a was decreased in Fig. 4B but increased in Fig. 4C.

Response: We thank the reviewer expert for constructive critiques.

We agree with the reviewer expert that the results have caused confusion, in which the addition of DN-OsMAPK6 exerted a different effect on the protein abundance of GW6a in different systems. To clear the confusion, we recently conducted the experiments for another 4-5 times (Figure 7). We thus chose the representative results for each of these systems (Figure 6). On the basis of these results, we favor a conclusion that the addition of DN-OsMAPK6 exerted a negative effect on the protein abundance of GW6a in different systems. We revised our manuscript accordingly.

Comments: 6. The difference in protein abundance between different lanes was subtle in some figures, such as Fig. 3C, Fig. 7E and Fig. 7F. I suggest the authors to confirm the results with more biological replicates.

Response: Thank you very much for the critical comments.

We also noticed that the difference in protein abundance between different lanes was subtle in the referred figures. Following the suggestion by the reviewer expert, we confirmed the results with additional three biological replicates as shown in Figures 10A-10C for each of the Figure 3C, Figure 7E and Figure 7F of our manuscript. We finally chose the representative results for each of these data (indicated by the red checkmarks).

Fig. 10 GW2 has a stronger ability than HDR3 and CLG1 to interact with GW6a.

Comments: 7. In Fig. 4H, why was the protein abundance of GW6a-Myc in the first lane (0 min) of each group so different?

Response: We thank the reviewer expert for the interesting comments.

The results shown in Figure 4H of our manuscript were derived from an *in vivo* protein degradation assay, wherein GW6a-Myc was individually co-expressed with the tag protein GFP, GFP fused CLG1, and CLG1^ΔR (deletion of the RING domain of CLG1) in tobacco leaves. Then, these co-expressed total proteins were extracted using the cell-free buffer and incubated for the indicated time and finally used for immunoblot assay. We have recognized that ubiquitylation of GW6a by CLG1 facilitates GW6a stabilization. This is the reason why the protein abundance of GW6a-Myc in the first lane (0 min) of each group was so different.

Comments: 8. The authors mentioned that "grain length and weight of OE-DN-OsMAPK6/OE-GW6a-Myc-2 did not exhibit additive genetic effects relative to those of OE-GW6a-Myc-2 and OE-DN-OsMAPK6, which were very similar to those of OE-GW6a-Myc-2 (Figures 5G, 5H)". The description is inaccurate and inconsistent with the Fig. 5G and 5H.

Response: We thank the reviewer expert for pointing out the inaccurate description.

We have corrected the inaccurate description in the new version of manuscript saying "As expected, we observed that grain length and weight of OE-DN-OsMAPK6/OE-GW6a-Myc-2 was between those of OE-GW6a-Myc-2 and OE-DN-OsMAPK6 (Figures 5G-5H), hinting that OE-GW6a-Myc could suppress the inhibitory effect on grain size and weight by Os-DN-OsMAPK6".

Comments: 9. Please explain why you chose two different systems for BiFC analysis. One in tobacco leaves (Figure 1C) and one in rice protoplast cells (Fig. 1F).

Response: Thank you very much for the comments.

The referred BiFC analyses in two different systems (i.e. tobacco leaves and rice protoplast cells) have been conducted by two different researchers (the first authors that contribute equally to this study) during different times. Nevertheless, the both systems have been recently adopted for BiFC analysis by our group and many other researchers (Guo et al., 2023; Ma et al., 2023; Shen et al., 2024; Xu et al., 2012; Yang et al., 2021; Yoshida et al., 2022).

Reviewer #3: Seed size and shape is important agronomic traits which determine the yield and quality in rice. GW6a, MAPK6, and CLG1 are reported as important seed size regulators via diverse mechanisms. However, the regulatory relationship among these three components is not known. In this manuscript, author identified two interaction partner of GW6a, a kinase MAPK6 and a E3 ligase CLG1. Then they demonstrate that MAPK6 and CLG can phosphorylate and ubiquitinate GW6a, respectively. Additionally, they mapped the phosphorylation site of GW6a by MAPK6. Interestingly, they found both MAPK6 and CLG promote the stabilization of GW6a. further, they show GW6a function downstream of MAPK6 and CLG via analyzing the seed size phenotype of double mutant. Finally, they identified several common regulated genes of GW6a, MAPK6, and CLG1 through RNA-Seq. Collectively, they suggest that two kinds of posttranslational modification of GW6a by MAPK6 and CLG1 might represent a novel seed size regulatory mechanism. The result is interesting, nevertheless, there have some points need to be clarified.

Response: We appreciate the reviewer expert for your positive comments.

Comments: 1. In this manus, author try to reveal the underlying regulatory mechanism among GW6a, MAPK6, and CLG1. However, they are three well-known seed shape regulator, which affect novelty of study. In addition, the intensity and depth of study is not enough, and some underlying deep-seated mechanism is not investigated and revealed, like whether phosphorylation of

GW6a by MAPK6 affect ubiquitination of GW6a?

Response: We thank the reviewer expert for the constructive critiques.

Fig. 11 Phosphorylation of GW6a by OsMAPK6 enhances its interaction with CLG1.

We agree with this reviewer expert that although GW6a, MAPK6, and CLG1 are three known seed-size regulators; however, as your early description that, the regulatory relationship among these three components is not known. In this study, we found that OsMAPK6 and CLG1 target GW6a for phosphorylation and ubiquitylation, respectively. However, it was unexpected that both of the two posttranslational modifications facilitate the stabilization of GW6a, which is different from known OsMAPK6 and CLG1-triggered effects. Furthermore, we showed that the *OsMAPK6-GW6a* and *CLG1-GW6a* axes were crucial and operated in a non-additive manner in the regulation of rice grain size. In addition, following the suggestion by the reviewer expert, we have recently performed pull-down assays to test whether phosphorylation of GW6a by OsMAPK6 affects its interaction with CLG1, by incubating the *E. coli*-produced GST fused GW6a, GW6a^{S142/T186A} and GW6a^{S142/T186D} with CLG1-Myc. Upon IP with GST, immunoblot assays using the anti-GST and anti-Myc antibodies suggested that compared with GST-GW6a, GST-GW6a^{S142/T186D} exhibits much stronger binding to CLG1-Myc, whereas GST-GW6a^{S142/T186A} has a much

weaker binding (Figure 11). Thus, phosphorylation of GW6a by OsMAPK6 facilitates its interaction with CLG1, and thus presumably has a positive effect on the CLG1 ubiquitination of GW6a. We discussed these results in the new version of our manuscript.

Comments: 2. Some results seems contradictory. They show overexpression of both NGW6a and CGW6a can increase seed size, implying that both NGW6a and CGW6a are functional. How NGW6a and CGW6a perform their biological function? Why overexpression of NGW6a and CGW6a show similar increased seed size phenotype? In addition, NGW6a, but not CGW6a, can be phosphorylated by MAPK6. These results might imply that phosphorylation of GW6a is not necessary for its biological function.

Response: We thank the reviewer expert for the insightful comments.

Our experimental data showed that both the transgenic rice plants with over-expression of HA tagged N-terminal part of *GW6a* (encoding amino acids 1-191; *OE-HA-nGW6a*) and *OE-HA-cGW6a* (*cGW6a* encodes amino acids 192-419) could increase grain size. However, *OE-HA-nGW6a* and *OE-HA-cGW6a* have increased grain length by over 14% and less than 5%, respectively (Supplemental Figure S2 of our manuscript). The reason for the different effects is due mainly to a fact that nGW6a contains a highly conserved GNAT domain for acetyl-CoA recognition and binding (Gao et al., 2021), and we inferred that cGW6a might have domain(s) for its interaction with other proteins involved in the regulation of rice grain size. In addition, our recent experimental results suggest that both the full-length GW6a and nGW6a could be phosphorylated by OsMAPK6 (Figure 1). We added these results and descriptions to the new version of our manuscript.

Comments: 3. It is known that mono-ubiquitination and poly-ubiquitination modification promote the stability and degradation of target protein,

respectively. In this manuscript, CLG1 promotes the stabilization of GW6a, but in most of the ubiquitination assays in this study, it looks like GW6a is poly-ubiquitinated by CLG1. So the ubiquitination of GW6a by CLG1 is mono-ubiquitination or poly-ubiquitination, which needs to be further investigated.

Response: We thank the reviewer expert for the interesting comments.

Fig. 12 Both CLG1 and GW2 polyubiquitylate and monoubiquitylate GW6a *in vitro*.

Ubiquitin (Ub) is linked to the substrate through an isopeptide bond between the C-terminal glycine residue of Ub and a lysine of the target protein (Thrower et al., 2000). A polyubiquitin chain is then formed by the addition of multiple ubiquitin monomers via an internal lysine (K) residue within Ub (Scherer et al., 1995). Ub has seven lysines at positions 6, 11, 27, 29, 33, 48, and 63 that can be utilized to form a polyubiquitin chain (Peng et al., 2003). We observed the ubiquitylation of GW6a by CLG1 and GW2 when we utilized the UbK0 variant (relative to the wild-type Ub), in which all seven lysine residues in Ub were changed into arginine, thus preventing the forming of polyubiquitylation chains (Ma et al., 2020). The following results suggested that both CLG1 and GW2 could monoubiquitylate GW6a, and the levels of GW6a monoubiquitylation

were markedly lower than those of a total of GW6a ubiquitylation (Figure 12). Thus, we reasoned that both CLG1 and GW2 could monoubiquitylate and polyubiquitylate GW6a *in vitro*. We added the results to the new version of our manuscript.

Comments: 4. In most the double mutant analysis, the overexpression of GW6a, CLG, and MAPK6 plants was used. As ectopic expression of gene will result in many unnatural phenotype, which affect the observed conclusion. For genetic analysis, the knockout mutant will be better than transgenic plants with ectopic expression.

Response: Thank you very much for the excellent suggestion.

We wholeheartedly agree with the reviewer expert that it is the best strategy to choose double knockout mutants for testing of genetic relationship between two genes. However, the implementation of this strategy was heavily impeded by the following existing fact. First, *OsMAPK6* is an essential gene for rice development and the heterozygous knockout mutant *osmapk6* plants failed to produce homozygous *osmapk6* seeds (Minkenberg et al., 2017). Second, *CLG1* positively regulates rice grain length; however, a knockout mutant of *CLG1* did not change grain length relative to the wild type presumably due to functional redundancy (Yang et al., 2021). Nevertheless, we found that knockout of *GW6a* (*cg-gw6a*) significantly reduced grain length, although it profoundly harmed grain filling; obviously, the grain size (length) of *cr-gw6a/CLG1p-CLG1-GFP-1* was significantly longer than that of *cr-gw6a*, but slightly shorter than that of *CLG1p-CLG1-GFP-1*, suggesting that *CLG1* and *GW6a* function in the same genetic pathway to regulate grain length (Figure 13). We added these results to our manuscript. In addition, we produced genetic crosses among over-expression of *GW6a* (*OE-GW6a-Myc-1*), *OE-CA-OsMAPK6*, *OE-CLG1-GFP*, *OE-DN-OsMAPK6*, and *OE-CLG1* Δ *R-GFP* for genetic analyses. This strategy has also been adopted recently by

our group and many other researches (Gao et al., 2021; Ma et al., 2023; Qiao et al., 2021; Shen et al., 2024; Sun et al., 2018; Yang et al., 2021).

Fig. 13 *CLG1* functions in a common genetic pathway with *GW6a* to regulate grain length

REFERENCES

Gao Q., Zhang N., Wang W.Q., Shen S.Y., Bai C. & Song X.J. The

ubiquitin-interacting motif-type ubiquitin receptor HDR3 interacts with and stabilizes the histone acetyltransferase GW6a to control the grain size in rice. *The Plant Cell* **33**: 3331 – 3347 (2021).

Guo J.P., Wang H.Y., Guan W., Guo Q., Wang J., Yang J., Peng Y.X., Shan

J.H., Gao M.Y., Shi S.J., Shangguan X.X., Liu B.F., Jing S.L., Zhang J., Xu

-
- C.X., Huang J., Rao W.W., Zheng X.H., Wu D., Zhou C., Du B., Cheng R.Z., Zhu L.L., Zhu Y.X., Walling L., Zhang Q.F., & He G.C. A tripartite rheostat controls self-regulated host plant resistance to insects. *Nature* **618**: 7991—807 (2023).
- Ma X.Y., Claus L., Leslie M., Tao K., Wu Z., Liu J., Yu X., Li B., Zhou J.G., Savatin D., Peng J., Tyler B., Heese A., Russinova E., He P. & Shan L.B. Ligand-induced monoubiquitination of BIK1 regulates plant immunity. *Nature* **581**: 199—203 (2020).
- Minkenberg B., Xie K.B., & Yang Y.N. Discovery of rice essential genes by characterizing a CRISPR-edited mutation of closely related rice MAP kinase genes. *The Plant J.* **89**: 636—648 (2017).
- Peng J., Schwartz D., Elias J.E., Thoreen C.C., Cheng D., Marsischky G., Roelofs J., Finley D. & Gygi S.P. A proteomics approach to understanding protein ubiquitination. *Nat. Biotechnol.* **21**: 921—926 (2003).
- Qiao J, Jiang H, Lin Y, Shang L, Wang M, Li D, Fu X, Geisler M, Qi Y, Gao Z, and Qian Q. A novel miR167a-OsARF6-OsAUX3 module regulates grain length and weight in rice. *Mol. Plant* **14**: 1683–1698 (2021).
- Scherer D.C., Brockman J.A., Chen Z., Maniatis T., & Ballard D.W. Signal-induced degradation of I κ B α requires site-specific ubiquitination. *Proc. Natl. Acad. Sci. USA* **92**: 11259—11263 (1995).
- Sun SY, Wang L, Mao HL, Shao L, Li XH, Xiao JH, Ouyang YD, and Zhang QF.

A G-protein pathway determines grain size in rice. *Nat. Commun.* **9**: 851–861 (2018).

Xu C., Wang Y.H., Yu Y.C., Duan J.B., Liao Z.G., Xiong G.S., Meng X.B., Liu G.F., Qian Q., & Li J.Y. Degradation of MONOCULM1 by APC/CTAD1 regulates rice tillering. *Nat Commun.* **3**: 750 (2012).

Yang W., Wu K., Wang B., Liu H., Guo S., Guo X., Luo W., Sun S., Ouyang Y., Fu X., Chong K., Zhang Q., and Xu Y. The RING E3 ligase CLG1 targets GS3 for degradation via the endosome pathway to determine grain size in rice. *Mol. Plant.* **14**: 1699–1713 (2021).

Yoshida H., Hirano K., Yano K., Wang F.M., Mori M., Kawamura M., Koketsu E., Hattori M., Ordonio R.L., Huang P., Yamamoto E., & Matsuoka M. Genome-wide association study identifies a gene responsible for temperature-dependent rice germination. *Nat Commun.* **13**: 5665 (2022).

REVIEWERS' COMMENTS

Reviewer #1 (Remarks to the Author):

My questions have been fully addressed, i have no more comments.

Reviewer #2 (Remarks to the Author):

This is a revised version of a manuscript that I had previously reviewed. The authors have performed additional experiments and constructively addressed all of my concerns, improving the manuscript significantly. Since several different proteins are involved in the post-translational modifications of GW6a, the authors have impressively demonstrated that GW2, a negative regulator of GW6a, competes with CLG1 and HDR3 to interact with GW6a to coordinate the regulation of grain size.

One minor suggestion: Please change the format of Figs. S9E and F to be consistent with the other four figures.